



# The Impact of Future Emission Policies on Tropospheric Ozone using a Parameterised Approach

Steven Turnock[1], Oliver Wild[2], Frank Dentener[3], Yanko Davila[4], Louisa Emmons[5], Johannes Flemming[6], Gerd Folberth[1], Daven Henze[4], Jan Jonson[7], Terry Keating[8], Sudo Kengo[9,10], Meiyun Lin[11,12], Marianne
Lund[13], Simone Tilmes[4], Fiona O'Connor[1]

[1]Met Office Hadley Centre, Exeter, UK
[2]Lancaster Environment Centre, Lancaster University, Lancaster, UK
[3]European Commission, Joint Research Centre, Ispra, Italy
[4]Department of Mechanical Engineering, University of Colorado, Boulder, Colorado, USA
[5]National Center for Atmospheric Research, Boulder, CO, USA
[6]European Centre for Medium-Range Weather Forecasts, Reading, UK
[7]EMEP MSC-W, Norwegian Meteorological Institute, Oslo, Norway
[8]U.S. Environmental Protection Agency, Washington D.C., USA
[9]Graduate School of Environmental Studies, Nagoya University, Nagoya, Japan
[10]Japan Agency for Marine-Earth Science and Technology (JAMSTEC), Yokohama, Kanagawa, Japan.
[11]Atmospheric and Oceanic Sciences, Princeton University, Princeton, USA
[12]NOAA Geophysical Fluid Dynamics Laboratory, Princeton, USA
[13]Center for International Climate and Environmental Research – Oslo (CICERO), Oslo, Norway

*Correspondence to*: Steven Turnock (steven.turnock@metoffice.gov.uk)

**Abstract.**

This study quantifies future changes in tropospheric ozone ($O_3$) using a simple parameterisation of source-receptor relationships based on simulations from a range of models participating in the Task Force on Hemispheric Transport of Air Pollutants (TF-HTAP) experiments. Surface and tropospheric $O_3$ changes are calculated globally and across 16 regions from perturbations in precursor emissions ($NO_X$, CO, VOCs) and methane ($CH_4$) abundance. A source attribution is provided for
each source region along with an estimate of uncertainty based on the spread of the results from the models. Tests against model simulations using HadGEM2-ES confirm that the approaches used within the parameterisation are valid. The $O_3$ response to changes in $CH_4$ abundance is slightly larger in TF-HTAP Phase 2 than in the TF-HTAP Phase 1 assessment (2010) and provides further evidence that controlling $CH_4$ is important for limiting future $O_3$ concentrations. Different treatments of chemistry and meteorology in models remains one of the largest uncertainties in calculating the $O_3$ response to perturbations
in $CH_4$ abundance and precursor emissions, particularly over the Middle East and South Asian regions. Emission changes for the future ECLIPSE scenarios and a subset of preliminary Shared Socio-economic Pathways (SSPs) indicate that surface $O_3$ concentrations will increase by 1 to 8 ppbv in 2050 across different regions. Source attribution analysis highlights the growing importance of $CH_4$ in the future under current legislation. A global tropospheric $O_3$ radiative forcing of +0.07 W m$^{-2}$ from 2010 to 2050 is predicted using the ECLIPSE scenarios and SSPs, based solely on changes in $CH_4$ abundance and tropospheric
$O_3$ precursor emissions and neglecting any influence of climate change. Current legislation is shown to be inadequate in limiting the future degradation of surface ozone air quality and enhancement of near-term climate warming. More stringent future emission controls provide a large reduction in both surface $O_3$ concentrations and $O_3$ radiative forcing. The parameterisation provides a simple tool to highlight the different impacts and associated uncertainties of local and hemispheric emission control strategies on both surface air quality and the near-term climate forcing by tropospheric $O_3$.





## 1 Introduction

Tropospheric ozone (O₃) is an air pollutant at both regional and global scales. It is harmful to human health (Brunekreef and Holgate, 2002; Jerrett et al., 2009; Turner et al., 2016; Malley et al., 2017), whilst also affecting climate (Myhre et al., 2013) and causing damage to natural and managed ecosystems (Fowler et al., 2009; United Nations Economic Commission for Europe (UNECE), 2016). Long-range transport of air pollutants and their precursors can degrade air quality at locations remote from their source region (Fiore et al., 2009). Predicting source-receptor relationships for O₃ is complex due to large natural background sources, formation of O₃ from local emissions, non-linear chemistry and inter-continental transport processes (TF-HTAP, 2010). In particular, it is uncertain how the interaction of local and regional emission controls with global changes (e.g. of methane and climate) could affect O₃ concentrations in the near-term future (2050s) (Jacob and Winner, 2009; Fiore et al., 2012; von Schneidemesser et al., 2015). This is evident from the wide range of modelled O₃ responses in future emission and climate scenarios (Kawase et al., 2011; Young et al., 2013; Kim et al., 2015). The setting and achieving of effective future emission control policies is therefore difficult, as a substantial proportion of O₃ comes from outside individual countries and regions.

Phase 1 of the Task Force on Hemispheric Transport of Air Pollutants (TF-HTAP1) (TF-HTAP, 2010) coordinated several sets of experiments using multiple models to study the source-receptor relationships from the intercontinental transport of O₃ and its precursors. It found that at least 30% of the total change in surface ozone concentration within a particular source region can be attributed to emission changes of similar magnitude that are external to the source region (TF-HTAP, 2010). This highlights the importance of source contributions outside the control of local/regional air pollutant policies, including those of stratospheric origin, natural sources and intercontinental transport. Changes in global methane (CH₄) concentrations are also an important contributor to baseline O₃ concentrations and are shown to be as important as changes in local source region emissions (TF-HTAP, 2010). Improving our understanding of the impact of anthropogenic emission changes on the source-receptor relationships arising from the intercontinental transport of tropospheric O₃ and its precursors will ultimately reduce the uncertainty in the impact of O₃ on air quality and climate, improving future predictions.

To predict how O₃ concentrations might respond to future changes in emissions, a simple parameterisation was developed based upon the surface O₃ response in different chemistry models contributing to TF-HTAP1 Wild et al., (2012). The surface O₃ response in these models was calculated from simulations with reductions in tropospheric O₃ precursor emissions across the four major northern hemisphere emission regions (Europe, North America, East Asia, and South Asia). The parameterisation using these results provided a fast and simple tool to predict future surface O₃ concentrations for the Intergovernmental Panel on Climate Change (IPCC) Representative Concentration Pathways (RCPs), highlighting the importance of future changes in emissions and CH₄ abundance for surface O₃ concentrations and quantifying the associated uncertainty.

A second phase of model experiments, TF-HTAP2, was initiated to extend the work from TF-HTAP1 and further consider the source-receptor relationships between regional emission reductions and air pollutants. Major advances in TF-HTAP2 include more policy-relevant source-receptor regions aligned to geo-political borders, a larger variety of idealised 20% emission reduction experiments, more recent (2008-2010) emission inventories that are consistent across all models and the use of new and updated models (Galmarini et al., 2017).

Here we improve and extend the parameterisation of Wild et al., (2012) by including additional information from TF-HTAP2 to refine the source-receptor relationships arising from emission changes, long range transport and surface O₃ formation. The parameterisation provides the contribution from local, remote and methane sources to the total surface O₃ response in each





emission scenario. The range of responses from the models contributing to the parameterisation provides an estimate of the uncertainty involved. The parameterisation is extended to estimate changes in tropospheric $O_3$ burden and its impact on $O_3$ radiative forcing. It is then used with the latest emission scenarios from ECLIPSE V5a (Klimont *et al.*, 2017; Klimont et al., *in prep.*) and the 6[th] Coupled Model Intercomparison Project (CMIP6) (Rao et al., 2017) to explore how source-receptor

relationships change in the future, informing the future direction of emission control policies. These predictions of changes in surface and tropospheric $O_3$ are based solely on changes in precursor emissions, as the parameterisation does not represent any impact from future changes in climate.

Section 2 of this paper describes the parameterisation and the updates from TF-HTAP1 to TF-HTAP2, including the extension

from surface $O_3$ to global tropospheric $O_3$ and its radiative forcing. Section 3 outlines the testing and validation of the parameterisation. A comparison is made to results from TF-HTAP1, highlighting changes in the $O_3$ response to changes in methane abundance. In section 4, the parameterisation is applied to the ECLIPSE V5a and CMIP6 emission scenarios to predict future surface $O_3$ concentrations over the period 2010 to 2050. Section 5 $O_3$ uses the same future emission scenarios to predict future tropospheric $O_3$ burden and radiative forcing. We conclude by suggesting how this approach could be used to inform

future emission policy in relation to $O_3$ concentrations.

## 2. Methods

### 2.1 Original Ozone Parameterisation

The parameterisation developed in this study is based on an earlier version developed for the TF-HTAP1 experiments by Wild et al., (2012). This simple parameterisation enabled the regional response in surface $O_3$ concentrations to be estimated based

on changes in precursor emissions and $CH_4$ abundance. The input for this parameterisation came from 14 different models that contributed to TF-HTAP1. All the models ran the same emission perturbation experiments (20% reduction in emissions of oxides of nitrogen ($NO_X$), carbon monoxide (CO), non-methane volatile organic compounds (NMVOCs) individually and all together) over the four major northern hemisphere source regions of Europe, North America, East Asia and South Asia. Additional experiments included global perturbations of emission precursors, as well as a 20% reduction in global $CH_4$

abundance. The multi-model responses from the 20% emission perturbation experiments are then scaled by the fractional emission changes from a given emission scenario over each source region. The monthly mean $O_3$ response ($\Delta O_3$) is the sum over each receptor region ($k$) of the scaled $O_3$ response from each model to the individual precursor species ($i$ - CO, $NO_X$ and NMVOCs) in each of the five source regions ($j$ - Europe, North America, East Asia, South Asia and rest of the world), including the response from the change in global $CH_4$ abundance (Eq. 1, reproduced from Wild et al., 2012):

$$\Delta O_3(k) = \sum_{i=1}^{3} \sum_{j=1}^{5} f_{ij} \Delta O_3(i,j,k) + f_m \Delta O_3(k) \qquad (1)$$

The emission scale factor $f_{ij}$ for each emission scenario is defined as the ratio of the fractional emission change ($\Delta E_{ij}/0.2 \times E_{ij}$) to the 20% emission reduction in the TF-HTAP1 simulations (Eq. 2). A similar scale factor for methane ($f_m$) is based on the

ratio of the change in the global abundance of $CH_4$ to that from the 20% reduced $CH_4$ simulation ($\Delta[CH_4]/0.2 \times [CH_4]$). Perturbations to emissions of CO, $NO_x$ and NMVOCs induce a long-term (decadal) change in tropospheric $O_3$ from the change in the oxidising capacity of the atmosphere (OH) and the $CH_4$ lifetime (Wild and Akimoto, 2001; Collins et al., 2002; Stevenson et al., 2004). The long-term impacts from 20% global emission reductions can reduce the $O_3$ response by 6-14% from $NO_x$ emission changes and increase the $O_3$ response by 16-21% from CO changes (West et al., 2007). This long-term response is

not accounted for in the simulations used here as $CH_4$ abundances are fixed.





Wild et al., (2012) found that this simple linear scaling relationship between emissions and surface $O_3$ was sufficient for small emissions perturbations, but that the relationship started to exhibit larger non-linear behaviour for larger perturbations, particularly for $NO_x$. The linear scaling factor is sufficient for the surface $O_3$ response from emission perturbations of CO and NMVOCs as non-linear behaviour from these precursors is small (Wu et al., (2009). To account for non-linear behaviour of

5  surface $O_3$ to $NO_x$ emission changes, the scale factor, $f$, is replaced with $g$, which has a quadratic dependency on $f$ (Eq. 2), and is based on additional simulations of surface $O_3$ response to larger emission perturbations undertaken in Wild et al., (2012).

For titration regimes, where a reduction in $NO_X$ emissions may lead to an increase in $O_3$, the surface $O_3$ response is limited (by a factor of $2f - g$ for emission reductions and for emission increases the linear scaling factor is used (Eq. 2). The spatial

10  extent and magnitude of titration regimes is assumed constant as it is based on model simulations from a single meteorological year.

The surface $O_3$ response to changes in global $CH_4$ abundances is also non-linear and showed similar behaviour to that due to $NO_x$ emissions. Therefore, $g$ in Eq. 2 is also used to represent the $O_3$ response to increases and decreases in $CH_4$ abundances.

In summary, the surface $O_3$ response to CO and NMVOC emission perturbations is represented by $f_{ij}$ and to changes in $CH_4$ abundances by $g_{ij}$ (Eq. 2). The representation of the surface $O_3$ response to $NO_x$ emissions is determined by the conditions specified in Eq. 2.

$$f_{ij} = \begin{cases} f_{ij} = \Delta E_{ij}/0.2 \times E_{ij} & \text{if } \Delta O_3(i,j,k) > 0 \text{ and } \Delta E_{ij} > 0 \\ 2f_{ij} - g_{ij} & \text{if } \Delta O_3(i,j,k) > 0 \text{ and } \Delta E_{ij} < 0 \\ g_{ij} & \text{otherwise (where } g = 0.95f + 0.05f^2) \end{cases} \qquad (2)$$

### 2.2 Phase 2 of TF-HTAP

A second phase of simulations has been undertaken as part of TF-HTAP to further study the transport of air pollutants and their impacts and to assess potential mitigation options (Galmarini et al., 2017). Phase 2 (TF-HTAP2) involved experiments using new and/or updated models that conducted idealised 20% perturbation simulations of $O_3$ emission precursors for different

source regions and source sectors over the years 2008 to 2010. A 20% emission perturbation was chosen to generate a sizeable response, whilst still being small enough to minimise non-linear chemistry effects. To determine the $O_3$ response to $CH_4$ changes, simulations increasing methane to 2121 ppbv (18%) and decreasing to 1562 ppbv (-13%) from a baseline of 1798 ppbv were undertaken in TF-HTAP-2. This range in $CH_4$ abundances was selected to encompass the uncertainty in $CH_4$ changes in 2030 from the 5[th] Coupled Model Intercomparison Project (CMIP5) scenarios of RCP8.5 and RCP2.6 (Galmarini

et al., 2017).

The source and receptor regions were updated to represent 16 new receptor regions (14 of which are also sources), aligned on geo-political and land/sea boundaries (Figure 1). Emission inventories (consistent across all models (Janssens-Maenhout et al., 2015)) and meteorology (driving data specific to individual models) were updated to consider the years 2008 to 2010 (the

focus of TF-HTAP1 was 2001). The Global Fire Emission Database version 3 (GFED3 – http://globalfiredata.org/) biomass burning (grassland and forest fires) emissions were recommended for TF-HTAP2 experiments, although some models selected other inventories. Individual modelling groups used their own information for other natural emission sources (e.g. biogenic VOCs, lightning $NO_x$), as many of these are based on internal model calculations and not externally prescribed datasets.

Priority in TF-HTAP-2 was placed on conducting a baseline simulation, a simulation with increased $CH_4$ concentrations and

seven regional simulations involving 20% reductions of all precursor emissions across the globe, North America, Europe, East





Asia, South Asia, Russia Belarus and Ukraine and the Middle East in the year 2010. A lower priority was given to emission perturbation experiments across the remaining source regions and experiments with individual emission perturbations and perturbations to individual source sectors. Models covered a consistent core set of 10-20 simulations and then conducted other experiments of their own choosing (see Galmarini et al., (2017) for a full list of models and experiments), resulting in sparse

coverage for many of the experiments. This contrasts with TF-HTAP1 where all models conducted the same set of 20% emission perturbation experiments covering all precursor emissions (individually and combined) and $CH_4$ across four source regions.

### 2.3 Improvements to the surface Ozone Parametric Model for TF-HTAP2

Differences in the experimental setup in TF-HTAP1 and TF-HTAP2 means that it is not straight forward to replace the

10 simulations underpinning the parameterisation of Wild et al., (2012) with those from TF-HTAP2. The larger number of simulations and fewer models involved preclude the development of a robust parameterisation based solely on TF-HTAP2 simulations. We therefore extend the existing parameterisation by including additional information from the new simulations in TF-HTAP2. To maintain a robust response over the major source regions of Europe, North America, East Asia and South Asia, results from the 14 models contributing to TF-HTAP1 over these regions were retained in the parameterisation. Results

from the models contributing to TF-HTAP2 were then incorporated, accounting for the different baseline year for emissions (2010 rather than 2001) and the change in size and number of source/receptor regions.

### 2.3.1 New Baseline Year

The baseline year used in the parameterisation was first adjusted from 2001 (TF-HTAP1) to 2010 (TF-HTAP2), to reflect changes in anthropogenic emissions between these years. It should be noted that the emission inventories used in TF-HTAP1

were not consistent between models, particularly for NMVOCs, and this partially contributed to the different $O_3$ responses (Fiore et al., 2009). In TF-HTAP2, the same anthropogenic emission inventory was used in all models to prevent uncertainty in anthropogenic emissions dominating the variability across models.

The parameterisation of Wild et al., (2012) was used to calculate new baseline $O_3$ concentrations in 2010 for use in the

25 improved parameterisation and for comparison to the TF-HTAP2 multi-model mean. To account for different $CH_4$ abundances the change between TF-HTAP1 and TF-HTAP2 was used. The mean fractional change in $NO_x$, CO and NMVOC emissions between 2000 and 2010 across the TF-HTAP1 source regions from two different emission inventories (MACCity (Granier et al., 2011) and EDGARv4.3.1 (Crippa et al., 2016)) was used due to the inconsistencies of the emissions inventories in TF-HTAP1 and TF-HTAP2 (Table 1). The MACCity and EDGAR inventories are internally consistent, enabling changes in

emissions for both future and historical time periods to be explored. Table1 shows that the emissions of $NO_x$, CO and NMVOC increased in Asia and decreased across Europe and North America over the period 2000 to 2010. Year 2000 was used as a consistent starting point for both emission inventories and can be considered equivalent of 2001 in representing changes to 2010.

The parameterised surface ozone response in 2010 across the original TF-HTAP1 source/receptor regions was compared to a multi-model ozone concentration in 2010 from the baseline simulations of seven TF-HTAP2 models that use the TF-HTAP2 emissions (Janssens-Maenhout et al., 2015). Table 2 shows that the $O_3$ concentrations from the parameterisation (H-P) are similar to the TF-HTAP2 multi-model mean values (H-2) over most of the receptor regions, and within the spread of individual model values (represented by one standard deviation). The large range and standard deviation in Table 2 highlights the large

spread in $O_3$ concentrations over the models in both sets of experiments (H-1 and H-2). The range in $O_3$ concentrations is much larger than the differences between the parameterised values and the TF-HTAP2 multi-model mean in 2010. This indicates



that the range of responses over the models dominates the uncertainty in O₃ concentrations and is much greater than differences due to the subset of models contributing to each study or from changing emissions over the period 2000 to 2010.

### 2.3.2 Source Region Adjustment

The original parameterisation was based on the continental-scale emission source regions defined in TF-HTAP1. To continue using these results in an improved parameterisation, the O₃ response fields were adjusted to represent the equivalent source regions in TF-HTAP2. The different regional definitions used within TF-HTAP1 and TF-HTAP2 experiments are shown in Figure 1 and are particularly large for Europe, where the TF-HTAP1 source region covers parts of five TF-HTAP2 source regions (Europe, Ocean, North Africa, Middle East and Russia Belarus and Ukraine). O₃ response fields from TF-HTAP1 models that formed the basis of the original parameterisation were adjusted to be more representative of the equivalent TF-HTAP2 source region.

No single model contributed experiments in both TF-HTAP1 and TF-HTAP2 to inform the adjustment of source regions. Therefore, 20% emission perturbation simulations were conducted with HadGEM2-ES (Collins et al., 2011; Martin et al., 2011), which contributed to TF-HTAP2 experiments, for the TF-HTAP1 source regions of Europe, North America, East Asia and South Asia. The ratio of the O₃ responses between the simulations using TF-HTAP1 and TF-HTAP2 source regions was then applied to the O₃ response fields from each of the TF-HTAP1 models used in the parameterisation of Wild et al., (2012). We assume that each model behaves in a similar way as HadGEM2-ES when the source regions are adjusted in this way. This generates an O₃ response field from emission perturbations within the equivalent TF-HTAP2 source regions of Europe, North America, East Asia and South Asia. The resulting O₃ parameterisation is based on a larger number of models (14 adjusted TF-HTAP1 models) than would have been available from using TF-HTAP2 simulations alone (7 TF-HTAP2 models), allowing for a larger diversity of model responses to represent the four major emission source regions.

### 2.3.3 Additions from TF-HTAP2

The O₃ responses from emission perturbations for the other ten TF-HTAP2 source regions were then used to augment the source region adjusted O₃ response fields from TF-HTAP-1. This extends the parameterisation to cover a much larger range of source regions (14 in total) than was previously possible. Table 3 lists the number of model simulations available for the TF-HTAP2 source regions over and above the four main source regions of Europe, North America, South Asia and East Asia, highlighting the sparseness of results for some of the TF-HTAP2 regions.

The monthly O₃ response fields from the additional ten TF-HTAP2 emission source regions were converted onto the same standard grid (1° x 1° in the horizontal, with 21 vertical levels based on regular pressure intervals from the surface at 1000 hPa to an upper level of 10 hPa) as used for the four source regions from the adjusted TF-HTAP1 models. In addition, the fields from the TF-HTAP1 models are based on the O₃ response to the individual emissions perturbations of NOₓ, CO and NMVOCs, whereas the regional emission perturbation simulations for TF-HTAP2 are based on all emission precursors together (due to the limited availability of results from regional individual precursor emission simulations in TF-HTAP2). To maintain consistency with the TF-HTAP1 parameterisation, the O₃ response for each TF-HTAP2 emission perturbation simulation is divided up to represent the response from individual emission precursors, as Wild et al., (2012) and Fiore et al., (2009) previously showed that O₃ responses from individual emission perturbations matched closely to that from combined emissions changes (within 2-7%). Therefore, the fractional contribution from individual emissions to the total O₃ response in the multi-model mean of TF-HTAP1 models is used to apportion the contribution from individual emissions in TF-HTAP2 simulations to the total O₃ response.



The $CH_4$ perturbation experiments in TF-HTAP2 were based on global changes of -13% and +18% to reflect the expected atmospheric abundance in RCP2.6 and RCP8.5, respectively. These were adjusted to the 20% reduction used in TF-HTAP1 using the parameterisation, allowing $O_3$ responses to $CH_4$ from the original 14 TF-HTAP1 models and the five TF-HTAP2 models that provided sufficient results to be combined.

### 2.3.4 Extension to Tropospheric Ozone

The parameterisation has been extended from the surface through the depth of the troposphere, enabling the calculation of the tropospheric $O_3$ burden. The three-dimensional monthly $O_3$ fields from the model simulations are interpolated onto 21 vertical levels at regularly spaced mid-level pressure intervals from 1000 hPa to 10 hPa. These $O_3$ fields were then used with the parameterisation to generate global and regional tropospheric $O_3$ burdens for each scenario, with the tropopause defined as an $O_3$ concentration of 150 ppbv (Prather et al., 2001). An $O_3$ radiative forcing is derived by using the tropospheric $O_3$ burden from the parameterisation and the relationship between radiative forcing and tropospheric column $O_3$ change based on multi-model ensemble mean results from the Atmospheric Chemistry and Climate Model Intercomparison Project (ACCMIP) (Stevenson et al., 2013). This relationship is provided as a two-dimensional global map, enabling regional and global $O_3$ radiative forcing to be calculated from the parameterisation.

### 3.0 Testing and Validation

The original parameterisation developed by Wild et al., (2012) was based on the surface $O_3$ response to 20% regional emission perturbations from TF-HTAP1 for 2001. We have updated this to reflect conditions in 2010, and have made significant improvements based on results from TF-HTAP2. The parameterisation has been extended to include the 14 sources regions in TF-HTAP2. Output is provided on a standard grid to facilitate the calculation of $O_3$ responses over any selected receptor regions. The parameterisation has been extended to generate three-dimensional $O_3$ fields that permit the calculation of tropospheric $O_3$ burden and $O_3$ radiative forcing for any scenario. To test and verify the improved parameterisation, additional simulations have been conducted with HadGEM2-ES, which are discussed in the following sections

### 3.1 Scaling Factors

We conducted experiments with HadGEM2-ES where all $O_3$ anthropogenic precursor emissions were reduced by 50% and 75% over Europe, to complement the existing 20% emission reduction scenarios performed as part of TF-HTAP2. Figure 2 shows a comparison of the annual and monthly surface $O_3$ response from the 20%, 50% and 75% European emission reduction simulations across Europe (a local receptor) and North America (remote receptor), using HadGEM2-ES and the parameterisation based on the $O_3$ response fields from HadGEM2-ES alone (a self-consistent test of the parameterisation). The largest errors of <1 ppbv occur over the source region (Fig. 2c), with smaller errors of <0.1 ppbv for the remote receptor region (Fig 2d). This small internal error between the parameterisation based on HadGEM2-ES and HadGEM2-ES simulations indicates that the parameterisation of $O_3$ is working well for emission changes at least as great as 50%. This is similar to the results of Wild et al., (2012) and indicates that the parameterisation performs well. Figure S1 compares the output of HadGEM2-ES simulations with the parameterisation based on $O_3$ response fields from multiple models. The magnitude of error is larger at ~2.0 ppbv over Europe and ~0.3 ppbv over North America for a 75% reduction. This highlights that the uncertainty in the parameterised $O_3$ response is dominated by the large spread in $O_3$ responses over the different models rather than by errors in the parameterisation.



### 3.2 Global Emission Perturbation

To further test the parameterisation, we compare the surface $O_3$ response from the parameterisation to a HadGEM2-ES model simulation using the ECLIPSE V5a current legislation scenario (CLE) in 2030 (see http://www.iiasa.ac.at/web/home/research/researchPrograms/air/Global_emissions.html, Klimont *et al.*, (2017) and Klimont

et al., (*in prep.*)). The ECLIPSE V5a emission scenarios provide future greenhouse gas and air pollutant emissions based on assumptions of energy use, economic growth and emission control policies for different anthropogenic emission sectors from the International Energy Authority (IEA). Three scenarios from ECLIPSE V5a are used in this study: Current Legislation (CLE) assumes future implementation of existing environmental legislation, Current Legislation with Climate policies (CLIM) is an energy and climate scenario targeting 2°C of climate warming in which air pollutants and $CH_4$ are reduced and Maximum

Technical Feasible Reduction (MTFR) is the introduction of maximum feasible available technology assuming no economic or technological constraints. Emissions of $O_3$ precursor species and $CH_4$ are available at decadal increments over the period 2010 to 2050 for each ECLIPSE scenario (MTFR is only available for 2030 and 2050). The $CH_4$ abundance was derived from the $CH_4$ emissions at decadal increments by using a simple box model that accounts for the sources and sinks of $CH_4$ and the feedbacks on its chemical lifetime following Holmes et al., (2013). Table 4 shows changes in annual $CH_4$ abundances and $NO_x$

emissions from the ECLIPSE scenario, with changes in CO and NMVOCs shown in Table S1 and S2 respectively. An ECLIPSE CLE 2030 scenario was generated by scaling the anthropogenic emissions in the TF-HTAP2 BASE scenario by the fractional emission changes in $NO_x$, CO and NMVOCs in CLE. A HadGEM2-ES simulation was performed using the change in emissions based on CLE for comparison to the parameterisation.

Fig 3. presents a comparison of monthly surface $O_3$ changes between 2010 and 2030 over the TF-HTAP2 regions for the ECLIPSE V5a CLE scenario from the HadGEM2-ES simulation and from the parameterisation (based solely on HadGEM2-ES model responses and based on responses from all models). This shows that the parameterisation is able to reproduce the magnitude and seasonality of surface $O_3$ changes over different regions when compared to the responses from a full global emission perturbation simulation. In particular, the parameterisation is able to reproduce the seasonality in $O_3$ across Europe

and North America, indicating that the adjustment to represent the new TF-HTAP2 sources regions is valid. Differences between the parameterisations highlight regions where HadGEM2-ES model responses differ from those of the multi-model mean. This is particularly evident for the Middle East, where there are differences of as much as 2 ppb. However, the parameterisation based on results of the HadGEM2-ES model alone agrees relatively well with the model simulation, as expected.

For South Asia, the parameterisation based on HadGEM2-ES and on the multi-model responses agrees well but differs substantially (in sign and magnitude) from the HadGEM2-ES simulated $O_3$ changes. The largest difference in surface $O_3$ concentrations of 5 ppbv between the model and the parameterisation occurs in the winter months (December, January, February), with differences in summer being much smaller (0.5 to 1 ppbv). Over the South Asian region the ECLIPSE CLE

emission scenario predicts a ~70% increase in $NO_X$ emissions by 2030 (Table 4). This large increase could lead to errors in the parameterised response due to the transition from $O_3$ production to titration, which the parameterisation is unable to represent well. However, it is able to represent the $O_3$ responses in the TF-HTAP2 models for a smaller emission change of 20% over South Asia (Table 5). The boundary layer mixing in HadGEM2-ES over South Asia (a region with challenging topography) has been shown to be insufficient, particularly in winter (Hayman et al., 2014; O'Connor et al., 2014), and a large

increase in $NO_x$ emissions could lead to a transition to $O_3$ titration over this region, accounting for the discrepancy in surface $O_3$ responses.





As the parameterisation of Wild et al., (2012) did not show a similar discrepancy over South Asia for large emission perturbations, a comparison has been made between the monthly surface $O_3$ response from HadGEM2-ES and the parameterisation across both the TF-HTAP1 and TF-HTAP2 definitions of South Asia in January and July (Figure 4). This shows that continental $O_3$ titration in January is less evident in the HadGEM2-ES simulation over the larger TF-HTAP1 South

Asia region, as it includes a large area of ocean. The TF-HTAP2 South Asia region is only continental and HadGEM2-ES shows the larger impact of $O_3$ titration over the continental region in January. The parameterisation and HadGEM2-ES $O_3$ responses agree much better over South Asia in July when there is less evidence of $O_3$ titration effects. The parameterisation, using only HadGEM2-ES as input, is not able to represent the $O_3$ response in HadGEM2-ES over TF-HTAP2 South Asia as it is based on a 20% emission reduction simulation of HadGEM-ES, where the extent of $O_3$ titration over the continental area is

small. Additional model simulations conducted with large emission increases over South Asia would be able valuable to further explore this issue, although none are currently available.

These results highlight that caution is needed when applying the parameterisation with emission changes larger than 50-60%, as noted previously in Wild et al., (2012). In particular, the shift into $O_3$ chemical titration regimes cannot be represented easily

in a simple parameterisation. For smaller emission changes, the parameterisation is shown to be relatively robust at representing monthly surface $O_3$ changes.

### 3.3 Comparison to HTAP-I

#### 3.3.1 CMIP5 Scenarios

We now use the improved parameterisation described above to explore how future predictions of regional surface $O_3$ for the

RCPs used in CMIP5 have changed since TF-HTAP1. The four RCPs assume different amounts of climate mitigation to reach a target anthropogenic radiative forcing in 2100: RCP2.6, RCP4.5, RCP6.0 and RCP8.5 (van Vuuren et al., 2011). Emissions of $O_3$ precursor species and $CH_4$ are available at decadal increments over the period 2010 to 2050 for each RCP. $CH_4$ emissions are converted to $CH_4$ abundances in each RCP using the MAGICC model which takes into account feedbacks on the $CH_4$ lifetime (Meinshausen et al., 2011). The parameterisation only accounts for the impact from changes in anthropogenic

emissions over the period 2010 to 2050 and does not account for changes in climate, but on this near-term timescale changes in $O_3$ are dominated by emission changes rather than climate effects (Fiore et al., 2012). There are large differences in global $CH_4$ abundances in the four scenarios, and this strongly influences the $O_3$ responses.

Figure 5 shows the change in surface $O_3$ across TF-HTAP2 regions for each of the RCPs. Surface $O_3$ decreases across most

regions in the majority of the scenarios as $O_3$ precursor emissions are reduced. The largest increases in surface $O_3$ occur over South Asia in RCP8.5 due to the expected increases in $O_3$ precursor emissions from 2010 to 2050, although we note that this effect may be exaggerated by the large increase in $NO_x$ emissions here in RCP4.5 and RCP8.5. Surface $O_3$ concentrations are predicted to increase over most regions in RCP8.5 with increases of 2 ppbv by 2050 over the Middle East and Southern Africa (Table S3). The results in Fig 5. across Europe, North America, South Asia, East Asia and globally are similar to those based

on TF-HTAP1 in Wild et al., (2012) (Fig. 5 and Table 6) but differ slightly in magnitude due to the change in the spatial extent of the individual source regions from TF-HTAP1 to TF-HTAP2. Additionally, the improved parameterisation provides $O_3$ changes for other regions that were not previously available, including the Middle East and Africa. This provides useful additional information on surface $O_3$ over these important regions under future emission change.

#### 3.3.2 Sensitivity of Ozone to Methane

The importance of controlling $CH_4$ to achieve future reductions in $O_3$ has been highlighted in earlier studies, along with the large uncertainty in the response of $O_3$ to $CH_4$ changes (Fiore et al., 2009; Wild et al., 2012). The inclusion of new models



provides an opportunity to assess whether the sensitivity of $O_3$ to $CH_4$ identified in TF-HTAP1 remain the same. Experiments with both increased (CH4INC) and decreased (CH4DEC) global abundance of $CH_4$ were conducted in TF-HTAP2. However, these experiments used an increase of 18% and a reduction of 13% to align with 2010 to 2030 changes in global $CH_4$ abundance under RCP8.5 and RCP2.6, in contrast to the 20% reduction used in TF-HTAP1.

Wild et al., (2012) found that a 20% increase in $CH_4$ abundance yielded an 11.4% smaller surface $O_3$ response than that from a 20% decrease in $CH_4$. For simplicity the parameterisation used the same scaling factor as for $NO_x$ emissions (Eq. 2 i.e. $g = 0.95f + 0.05f^2$), which represents a 10% smaller response for successive emission increases. The two TF-HTAP2 models that contributed results to both CH4DEC and CH4INC simulations allow us to check the expression used here. We find a

10     slightly larger sensitivity, with both models yielding a 12.6% smaller surface $O_3$ response for an increase in $CH_4$ than a decrease (Eq. 3). Since this $O_3$ response to $CH_4$ in TF-HTAP2 is comparable to that from TF-HTAP1, for simplicity and consistency we chose to retain the same scaling factor for both $NO_x$ and $CH_4$ (Eq. 2).

$$g = 0.937f + 0.063f^2 \tag{3}$$

To enable a direct comparison with TF-HTAP1 results, the $O_3$ response from the CH4DEC and CH4INC experiments in TF-HTAP2 are scaled to represent the response from a 20% reduction in $CH_4$ abundances, using Eq. 3. An adjustment factor is calculated based on the global mean difference between the TF-HTAP2 $O_3$ response in each experiment and that of an equivalent 20% reduction in $CH_4$ abundance (calculated using Eq. 3), resulting in a factor of 1.557 for CH4DEC and -1.256

for CH4INC. The global mean $O_3$ responses from CH4DEC (-0.69 ± 0.01 ppbv, 2 models) and CH4INC (0.81 ± 0.14 ppbv, 7 models) are adjusted to generate the equivalent $O_3$ responses to a 20% reduction in $CH_4$ abundance, which are used in the parameterisation (-1.05 ± 0.12 ppbv). This response is ~14% larger globally than that in TF-HTAP1 (-0.90 ± 0.14 ppbv, 14 models), highlighting a slightly increased sensitivity of $O_3$ to $CH_4$.

To explore the differences between TF-HTAP1 and TF-HTAP2 models the $CH_4$ lifetime and feedback factor for each TF-HTAP2 model (where data is available) can be calculated in accordance with Fiore et al., (2009). The feedback factor is the ratio of the atmospheric response (or perturbation) time to global atmospheric lifetime and describes how the atmospheric $CH_4$ abundance responds to a perturbation in $CH_4$ emissions e.g. a feedback factor of 1.25 means that a 1% increase in emissions would ultimately generate a 1.25% increase in $CH_4$ concentrations (Fiore et al., 2009). The feedback factors can be used in

conjunction with $CH_4$ emission changes for a region, to relate the $O_3$ response from the reduction in $CH_4$ abundance in TF-HTAP scenarios to that equivalent from emissions, taking into account both the long-term and short response of emissions on $O_3$ (Fiore et al., 2009). Table 7 summarises the calculated $CH_4$ lifetime and feedback factors for the two TF-HTAP2 models that have provided the appropriate fields. These two models show slightly shorter methane lifetimes and a higher feedback factor (F) than the TF-HTAP1 mean values. This suggests that the sensitivity of $O_3$ to changes in $CH_4$ in the two TF-HTAP2

models is slightly larger than the TF-HTAP1 multi-model mean. The increased feedback factor also indicates that a slightly larger reduction in methane emissions is required to achieve a comparable reduction in $O_3$ concentrations.

Overall, the sensitivity of $O_3$ to a change in $CH_4$ abundance is slightly larger in the two TF-HTAP2 models considered here than in TF-HTAP1 models, but still within the range of the TF-HTAP1 multi-model ensemble. The results from TF-HTAP2

will not significantly change any conclusions from TF-HTAP1 but suggests that the previous $O_3$ changes estimated from TF-HTAP1 are conservative. The $O_3$ response to $CH_4$ remains one of the most important processes to understand for controlling future $O_3$ concentrations.




## 4. Future Surface Ozone Predictions

### 4.1 Surface Ozone under ECLIPSEv5a Emissions

The parameterised approach is used with the ECLIPSE v5a emission scenarios described above to determine regional changes in future surface $O_3$ concentrations. Surface $O_3$ concentrations for the CLE (current legislation) scenario are predicted to
increase from 2010 to 2050 across all regions (Figure 6). Annual mean surface $O_3$ concentrations increase by 4 to 8 ppbv across the South Asia and Middle East regions due to the large increases expected in $NO_x$ emissions (Table 4), although there is substantial uncertainty in the parameterisation over these regions. Surface $O_3$ concentrations over Europe and North America in 2050 are similar to those in 2010, even though their regional $NO_x$ emissions decrease by ~50%. The contributions of different sources to the total surface $O_3$ change has been analysed for each source region (Figures S2 to S13). Results for Europe (Figure.
7) and South Asia (Figure. 8) are shown here, as these regions experience contrasting changes in surface $O_3$. Across Europe, surface $O_3$ from local and remote (mainly North American) sources is reduced in response to emission decreases, and the contribution from $CH_4$ increases by 1.6 ppbv in 2050 (Fig. 7). The increase in global $CH_4$ abundance in the CLE scenario increases surface $O_3$ over Europe, offsetting the reduction in $O_3$ from local and remote sources. This contrasts strongly with South Asia where local sources dominate the total $O_3$ response. This demonstrates how different local and hemispheric
emission control strategies are needed in different regions.

For the CLIM (climate policies on current legislation) scenario, annual mean surface $O_3$ concentrations in 2050 decrease slightly or stay at 2010 concentrations due to reductions in anthropogenic emissions and control of $CH_4$ emissions leading to a decrease in its abundance (Table 4). The source contribution analysis for Europe (Fig. 7) and South Asia (Fig. 8) shows that
$CH_4$ contributes much less to the total surface $O_3$ change under this scenario than CLE. For South Asia, there is also a reduction in the contribution from local sources to surface $O_3$. Under CLIM, remote sources start to dominate the contribution to European surface $O_3$ changes in 2050, increasing to -1.3 ppbv. However, across South Asia the contribution from local sources (+3.2 ppbv) is greater than from remote sources (-1.4 ppbv) in 2050, reflecting the importance of local emissions in this region. The contribution of $CH_4$ sources to the total surface $O_3$ response is smaller in CLIM due to the targeting of $CH_4$ for climate
mitigation purposes. The implementation of these climate policy measures shifts the dominant factor driving future $O_3$ changes within a receptor region towards extra-regional sources.

The MTFR scenario (maximum technically feasible reduction) considers large reductions in emissions (Table 4) and consequently predicts reductions in surface $O_3$ concentrations of up to 9 ppbv by 2050. Reductions of surface $O_3$ in Europe
(Fig. 7) are dominated by changes in remote sources, although changes in $CH_4$ become increasingly important by 2050. For South Asia, the surface $O_3$ response is dominated by changes to local and remote emission sources. This highlights that achieving decreases in surface $O_3$ concentrations from the maximum feasible emissions reductions depends not only on local emission policies but on reducing emissions across other regions too.

### 4.2 Surface Ozone under CMIP6 Emissions

We provide an initial assessment of surface $O_3$ changes from a subset of the preliminary emission scenarios developed for the CMIP6 project (https://tntcat.iiasa.ac.at/SspDb; Rao *et al.*, 2017) based on shared socio-economic pathways (SSPs). Five baseline SSPs are defined (SSP1-5) based on different combinations of future social, economic and environmental development trends over centennial timescales (O'Neill et al., 2014). Different climate targets, defined in terms of anthropogenic radiative forcing by 2100, are combined with the baseline SSPs to develop future scenarios for climate
mitigation, including additional assumptions on international co-operation, timing of mitigation and extent of fragmentation between low and high income economies (van Vuuren et al., 2014; Riahi et al., 2017). Scenarios of strong, medium and weak





future air pollutant emission pathways are mapped onto the SSPs and represent differing targets for pollution control, the speed at which developing countries implement strict controls and the pathways to control technologies (Rao et al., 2017). Increasingly stringent air pollutant emission controls are assumed to occur with rising income levels because of the increased focus on human health effects and the declining costs of control technology. SSP2 is a medium pollution control scenario that
follows current trajectories of increasing levels of regulation. SSP1 and SSP5 are strong control scenarios where pollution targets become increasingly strict. A weak pollution control scenario is adopted in SSP3 and SSP4 where the implementation of future controls are delayed (Rao et al., 2017).

We select three preliminary SSPs to represent scenarios of business as usual (SSP3 BASE), middle of the road (SSP2 60) and
10 enhanced mitigation (SSP1 26). The SSP2 60 and SSP1 26 scenarios have climate mitigation targets of 6.0 and 2.6 W m$^{-2}$ in 2100 applied to them. Currently, air pollutant emissions for each SSP are available globally and across five world regions from https://tntcat.iiasa.ac.at/SspDb/. The air pollutant emissions for each region have been mapped onto the equivalent TF-HTAP2 source regions and the grouping of regions is shown in Table 8 along with the percentage change in global $CH_4$ abundance and $NO_x$ emissions over the period 2010 to 2050. The relative changes in CO and NMVOCS emissions are shown in Table S4
and S5. Gridded versions of these emission scenarios will be made available in due course (K. Riahi, personal communication, 2017), which will allow a more accurate evaluation of the impacts arising from these scenarios.

Surface $O_3$ concentrations increase across all regions in 2050 for the SSP3 BASE scenario (Figure 9). Europe, North America and East Asia show an increase in surface $O_3$ of 1 to 3 ppbv, a larger response than in the ECLIPSE CLE scenario. Smaller
increases in surface $O_3$ are predicted over the Middle East (~3 ppbv) and South Asia (~5 ppbv) compared to CLE. Methane dominates the total surface $O_3$ response over Europe in SSP3 BASE, with small contributions from local and remote emission sources over the period 2010 to 2050 (Fig. 10). Local emissions are the main contribution to $O_3$ changes over South Asia, with a slightly larger influence from $CH_4$ than in CLE (Fig. 11).

For SSP2 60 (middle of the road scenario), surface $O_3$ concentrations reduce slightly by 2050 and to a greater extent than in ECLIPSE CLIM due to the larger reductions in $NO_x$ emissions and global $CH_4$ abundances (Table 4 and 9). Over South Asia, $NO_x$ emissions in SSP2 60 decrease by 22% from 2010 to 2050, with a corresponding $O_3$ change of -3 ppbv, compared to CLIM where $NO_x$ emissions increase by 66% and the corresponding $O_3$ change is +3 ppbv. However, this difference could arise from using preliminary SSP emissions based on five large world regions, where emission changes in South Asia and
nearby regions such as East Asia are combined together. For Europe and South Asia the source contributions for each region (Fig. 10 and 11) are similar to those in CLIM. Remote sources are more important under this intermediate climate mitigation scenario, with local emissions sources becoming more important by 2050 over South Asia.

Large reductions in surface $O_3$ concentrations are predicted across all regions in the strong mitigation scenario (SSP1 26) (Fig.
9). The improvements in $O_3$ concentrations are less than predicted under the ECLIPSE MTFR due to a smaller reduction in $NO_x$ emissions. Northern mid-latitude regions show reductions in surface $O_3$ concentrations of up to 6 ppbv under SSP1 26, similar to MTFR. Over South Asia, surface $O_3$ is predicted to be reduced by up to 7 ppbv, which is less than under MTFR. The source contributions for both Europe (Fig. 10) and South Asia (Fig. 11) are similar to MTFR with the importance of remote sources and the increasing importance of $CH_4$ by 2050 evident over Europe. Over South Asia, the increasing importance
of local and $CH_4$ sources is clear by 2050.

This analysis of preliminary CMIP6 emission scenarios highlights the large range of future regional surface $O_3$ responses that are possible depending on the climate and air pollutant policies applied. The assumptions within each of the future SSPs,





particularly for CH$_4$, results in different sources dominating the contribution to the total surface O$_3$ response. Uncertainties in the assumed growth rate of CH$_4$ under the two current legislation scenarios (CLE and SSP3 BASE) result in a 1 ppbv difference in surface O$_3$ over Europe and North America, highlighting the importance for future air quality of reducing CH$_4$ on a global scale. The CMIP6 scenarios allow a larger range of pathways to be explored than were available in ECLIPSE or the CMIP5

RCPs, including those of strong, medium and weak policies on air pollutants and climate change. The parameterisation can be used to provide a rapid assessment of the impact of differing policy measures on surface O$_3$ concentrations across different regions, along with a clear source attribution. This can ultimately inform selection of policies that are most beneficial to future air quality.

## 5. Future Tropospheric Ozone Burden and Radiative Forcing

As discussed in section 2.3, the parameterisation has been extended to generate three-dimensional O$_3$ distributions throughout the troposphere, using a tropopause defined as a O$_3$ concentration of 150 ppbv (Prather et al., 2001). Tropospheric O$_3$ column burdens are calculated in each grid cell for each emission scenario. These are used to infer changes in O$_3$ radiative forcing by using the relationship between radiative forcing and tropospheric column O$_3$ (W m$^{-2}$ DU$^{-1}$) and its spatial variation with latitude and longitude from the ACCMIP multi-model ensemble (Stevenson et al., 2013). Tropospheric O$_3$ burdens and O$_3$ radiative

forcings are calculated for the CMIP5 RCPs to evaluate the parameterisation against values from the ACCMIP multi-model study (Stevenson et al., 2013). Additionally, future projections of O$_3$ radiative forcing are made for the ECLIPSE and CMIP6 SSPs.

The change in tropospheric O$_3$ burden for the ECLIPSE CLE scenario in 2030 from the parameterisation was evaluated against

the change from the equivalent HadGEM2-ES simulation, a self-consistent test. The parameterisation produced a change in global tropospheric O3 burden of -0.9 Tg which compares well with the -1.0 Tg change from HadGEM2-ES simulations. In comparison to the ACCMIP multi-model mean (Stevenson et al., 2013), the parameterisation produces changes in the global O$_3$ burden and O$_3$ radiative forcing from 2000 to 2030 for RCP2.6 and RCP6.0 (Table 9) relatively well but underestimates the magnitude of change in RCP4.5 and RCP8.5 (although it remains within the range of ACCMIP model responses). The

parameterisation also underestimates the changes in global O$_3$ burden and radiative forcing between 1980 and 2000 compared to the ACCMIP multi-model mean. The parameterised response compares reasonably well for smaller emission changes but is unable to reproduce fully the larger changes in the high emission (low mitigation) scenario (RCP8.5), as it does not include natural emissions or represent impacts on O$_3$ from changes in climate that are contained within the ACCMIP models. The net impact of climate change on global tropospheric O$_3$ radiative forcing was estimated from the ACCMIP multi-model ensemble

to be between -20 to -30 mW m$^{-2}$ (a negative feedback) (Stevenson et al., 2013).

A global O$_3$ radiative forcing of +0.05 to +0.08 W m$^{-2}$ in 2050, relative to 2010, is estimated under the low mitigation scenarios (RCP8.5, CLE and SSP3 BASE) (Figure 12). The intermediate mitigation scenarios of RCP4.5, RCP6.0 and SSP2 60 show an O$_3$ radiative forcing of 0 to -0.04 W m$^{-2}$ in 2050, with almost no change under CLIM. The more stringent mitigation scenarios

(RCP2.6, MTFR and SSP1 26) exhibit an O$_3$ radiative forcing of between -0.07 and -0.15 W m$^{-2}$ by 2050. The parameterisation is able to predict the wide range of impacts that climate and air quality policies could have on short-term climate forcing from O$_3$. It can be used as a rapid screening tool to select the most appropriate climate scenarios to explore further in full model simulations that can provide more detailed predictions. The current business-as-usual scenarios for CMIP5, ECLIPSE and CMIP6 increase the climate forcing of O$_3$ by approximately 0.06 W m$^{-2}$ in 2050, whereas the strong mitigation scenarios have

a larger effect in reducing the near-term climate forcing of O$_3$ by about 0.10 W m$^{-2}$.





The parameterisation generates gridded changes in the tropospheric $O_3$ column burden and radiative forcing, which can be used to calculate changes over different regions. Figures 13 and 14 show that the largest relative changes in $O_3$ burden for the ECLIPSE scenarios occur over the Middle East, South Asia and South East Asia (> 10%), with a corresponding larger impact on $O_3$ radiative forcing (-0.3 W m$^{-2}$ in MTFR). Smaller relative changes in the tropospheric $O_3$ burden are found for CLE over

Europe and North America. For MTFR a 15% reduction in $O_3$ burden is predicted over Europe and North America, similar to that over South Asia, but the change in $O_3$ radiative forcing is not as large (-0.2 W m$^{-2}$ compared to -0.3 W m$^{-2}$ over South Asia). The parameterisation allows the regional near-term climate implications (in terms of $O_3$ radiative forcing) from future emissions changes to be explored under different air quality and climate policy scenarios. It also highlights the wide range of near-term climate forcing that is possible over particular regions from future emission policies.

**6. Conclusions**

In this study, we describe improvements and extensions to a simple parameterisation of regional surface $O_3$ responses to changes in precursor emissions and $CH_4$ abundances based on multiple models. We incorporate results from phase 2 of the Hemispheric Transport of Air Pollutants project to create an enhanced parameterisation that includes new models, a greater number of source regions, a new baseline of 2010 and an extension to three dimensions to represent $O_3$ changes throughout

the troposphere. These improvements allow impacts on surface $O_3$ concentrations and the near-term $O_3$ radiative forcing to be calculated from different emission scenarios. Model simulations using HadGEM2-ES confirm the validity of the parameterisation and adjustments made here. There is a slight increase in the response of $O_3$ to $CH_4$ for the TF-HTAP2 models, resulting in a slightly higher sensitivity of $O_3$ to $CH_4$ changes. The extent of the difference varies on a regional basis, but is within the range of model responses in TF-HTAP1.

The parameterisation is shown to perform well under most conditions, although there are larger uncertainties for future surface $O_3$ responses over South Asia where changes in emissions are particularly large. Emission changes from the RCPs are used with the parameterisation and it predicts similar changes in surface $O_3$ concentrations to those from the original parameterisation (Wild et al., 2012), although now across a larger number of source/receptor regions. Tropospheric $O_3$ burden

and $O_3$ radiative forcing calculated using the parameterisation compare well with the ACCMIP multi-model mean values for intermediate emission scenarios (RCP2.6 and RCP6.0) in 2030 but underestimate the large changes in RCP8.5 due to the neglect of climate induced changes within the parameterisation. The parameterised approach permits rapid assessment of the impact of future emission changes over 14 source regions and associated uncertainties on both surface and tropospheric $O_3$ concentrations, and allows identification of the differing contributions of local, remote and $CH_4$ sources to the $O_3$ response.

This enables quantification of the impacts of future air quality and climate emission policies on surface air quality and near-term climate forcing by $O_3$.

Applying future emissions from ECLIPSEv5 and the preliminary SSPs, we show that annual mean surface $O_3$ concentrations are likely to increase across most world regions by 2050 under current legislation scenarios, with large increases of 4 to 8 ppbv

over the Middle East and South Asia. These changes in $O_3$ concentrations are driven mainly by local emissions and changes in global $CH_4$ abundance. This demonstrates that current legislation is inadequate in preventing future increases in surface $O_3$ concentrations across the world. Implementing energy related climate policies on top of current legislation maintains future surface $O_3$ concentrations at or slightly below 2010 concentrations, counteracting the increases that occur under current legislation. This is achieved mainly through reductions in $CH_4$, highlighting the importance of controlling $CH_4$ in limiting

future changes in $O_3$ concentrations, as shown in Wild et al., (2012). Policies that have stringent emission controls lead to





substantial reductions in surface $O_3$ concentrations across all world regions of up to 8 ppbv and could potentially provide large beneficial impacts.

A global $O_3$ radiative forcing of +0.07 W m$^{-2}$ is predicted by 2050 (relative to 2010) under the current legislation scenarios of
the SSPs and ECLIPSE. There is a large and diverse regional response in $O_3$ radiative forcing with some regions e.g. Middle East and South Asia more sensitive to changes in emissions than others, and these show a large positive $O_3$ radiative forcing under current legislation. However, application of aggressive emission mitigation measures leads to large reductions in $O_3$ radiative forcing (-0.10 W m$^{-2}$), lessening the near-term impact on climate.

The new parameterisation provides a valuable assessment tool to evaluate the impact of future emission policies on both surface air quality and near-term climate forcing from $O_3$. It also provides a full source attribution along with a simple measure of uncertainty, given by the spread of the multi-model responses that reflect different transport and chemistry processes in models. Whilst not replacing full chemistry simulations it provides a quick way of assessing where to target future modelling efforts. However, these $O_3$ responses are based on changes to anthropogenic emissions only, with no account taken of the
impact on $O_3$ and/or its natural precursor emissions due to future changes in chemistry or climate. The parameterisation could be extended further by including a feedback factor to take some account of the impact of future climate change on $O_3$. Additional improvements could include coupling the output to an offline radiation model to enable improved calculation of $O_3$ radiative forcing, using $O_3$ fields from the parameterisation within a land surface model to assess the impacts of $O_3$ on vegetation and the carbon cycle or with $O_3$ dose-response functions to calculate impacts on human health.

**Acknowledgments**

Steven Turnock, Fiona O'Connor and Gerd Folberth would like to acknowledge the BEIS Met Office Hadley Centre Climate Program (GA01101). Steven Turnock also acknowledges the UK-China Research and Innovation Partnership Fund through the Met Office Climate Science for Service Partnership (CSSP) China as part of the Newton Fund. Simone Tilmes would like to acknowledge high-performance computing support from Yellowstone (ark:/85065/d7wd3xhc) provided by NCAR's
Computational and Information Systems Laboratory, sponsored by the National Science Foundation. The CESM project is supported by the National Science Foundation and the Office of Science (BER) of the U. S. Department of Energy. . Frank Dentener acknowledges support from the AMITO2 Administrative Arrangement with DG ENV. The authors acknowledge access to the preliminary version of the SSP air pollution database hosted at the International Institute for Applied Systems Analysis.

*Competing Interests*. The authors declare that they have no conflict of interest.

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





**Table 1. Summary of annual mean NOₓ, CO and NMVOC emissions changes (%) between 2000 and 2010 over TF-HTAP-1 source regions (values are mean differences from the MACCity and EDGARv4.3.1 emission inventories)**

|  | Annual total relative (%) emission change between 2000 and 2010 | | | | | |
|---|---|---|---|---|---|---|
|  | **Global** | **Europe** | **North America** | **South Asia** | **East Asia** | **Rest of World** |
| NOₓ | 9.5 | -8.4 | -25.0 | 49.8 | 42.1 | 13.0 |
| CO | -1.2 | -27.1 | -47.1 | 18.8 | 15.6 | 9.0 |
| NMVOCs | 5.2 | -9.7 | -31.2 | 32.1 | 24.8 | 10.0 |



**Table 2. Summary of multi-model annual mean surface ozone values from all TF-HTAP1 models in 2001 (H-1), the parameterisation of TF-HTAP1 models scaled for emissions in 2010 (H-P) and all TF-HTAP2 models in 2010 (H-2) (seven models contributing).**

| | Ozone Concentrations (ppbv) | | | | | | | | | | | | | | |
|---|---|---|---|---|---|---|---|---|---|---|---|---|---|---|---|
| | Global | | | Europe | | | North America | | | South Asia | | | East Asia | | |
| | H-1 | H-P | H-2 | H-1 | H-P | H-2 | H-1 | H-P | H-2 | H-1 | H-P | H-2 | H-1 | H-P | H-2 |
| Min | 21.2 | 21.5 | 23.0 | 30.2 | 30.1 | 29.9 | 29.4 | 28.3 | 29.7 | 35.2 | 37.6 | 35.3 | 28.9 | 30.8 | 31.7 |
| Mean | 27.4 | 27.2 | 26.4 | 37.4 | 36.9 | 35.8 | 35.8 | 34.9 | 35.1 | 40.1 | 42.4 | 40.7 | 35.5 | 37.2 | 35.5 |
| Max | 32.0 | 30.0 | 32.3 | 42.8 | 42.4 | 42.0 | 40.8 | 39.7 | 41.2 | 44.8 | 48.0 | 50.7 | 38.9 | 40.7 | 41.3 |
| Standard Deviation | 2.94 | 2.71 | 3.33 | 3.84 | 3.79 | 4.45 | 3.56 | 3.54 | 3.85 | 3.73 | 3.80 | 3.35 | 2.92 | 2.91 | 5.27 |

5  NB - The TF-HTAP2 models used to provide the 2010 ozone concentrations are CAMchem, Chaser_re1, Chaser_t106, C-IFS, GEOS-Chem adjoint, HadGEM2-ES, Oslo-CTM



**Table 3. Models contributing to each of the TF-HTAP2 emission perturbation experiments, in addition to those for the source regions of Europe, North America, South Asia and East Asia**

| TF-HTAP2 Model | TF-HTAP2 Experiment | | | | | | | | | | | |
| --- | --- | --- | --- | --- | --- | --- | --- | --- | --- | --- | --- | --- |
| | CH4INC | CH4DEC | MDE | RBU | NAF | SAF | MCA | SAM | SEA | CAS | PAN | OCN |
| GFDL-AM3 (Lin et al., 2012) | X | | | | | | | | | | | |
| C-IFS (Flemming et al., 2015) | X | | | | | | | | | | | |
| CAM-Chem (Tilmes et al., 2016) | | | X | X | | | | | | | | |
| CHASER_re1 (Sudo et al., 2002) | X | | X | X | | | | | | | | |
| CHASER_t106 (Sudo et al., 2002) | X | | | | | | | | | | | |
| EMEP_rv4.8 (Simpson et al., 2012) | X | | X | X | X | | | | | | | X |
| GEOS-Chem (Henze et al., 2007). | | | X | X | | | | | | | | |
| HadGEM2-ES (Collins et al., 2011) | X | X | X | X | X | X | X | X | X | X | X | |
| OsloCTM3_v2 (Søvde et al., 2012) | X | X | X | X | | | X | X | | | | |
| Total Number of Models | 7 | 2 | 6 | 6 | 2 | 1 | 2 | 2 | 1 | 1 | 1 | 1 |

5   MDE – Middle East, RBU – Russia Belarus Ukraine, NAF – North Africa, SAF – Southern (Sub-Saharan/Sahel) Africa, MCA – Mexico and Central America, SAM – South America, SEA – South East Asia, CAS – Central Asia, PAN – Pacific Australia and New Zealand, OCN – Ocean (for region definitions see Koffi et al., 2016)





**Table 4.** Percentage change in global CH₄ abundance and global and regional annual NOₓ emissions relative to 2010 over each TF-HTAP2 region for the different ECLIPSE V5a emission scenarios (CLE, CLIM and MTFR). MTFR scenarios are only available for 2030 and 2050.

| TF-HTAP2 Region | Annual total emission change (%) from 2010 | | | | | | | | | |
| | CLE | | | | CLIM | | | | MTFR | |
| | 2020 | 2030 | 2040 | 2050 | 2020 | 2030 | 2040 | 2050 | 2030 | 2050 |
| --- | --- | --- | --- | --- | --- | --- | --- | --- | --- | --- |
| Global CH₄ Abundance | 4 | 12 | 21 | 32 | 3 | 8 | 11 | 13 | -9 | -21 |
| Global NOₓ | -7 | -6 | 6 | 19 | -17 | -27 | -26 | -24 | -88 | -86 |
| *Regional NOₓ Emissions* | | | | | | | | | | |
| Central America | 13 | 11 | 21 | 30 | 1 | -16 | -16 | -11 | -46 | -79 |
| Central Asia | 10 | 15 | 18 | 26 | -5 | -16 | -26 | -32 | -57 | -80 |
| East Asia | -14 | -16 | -8 | -3 | -16 | -27 | -25 | -24 | -50 | -61 |
| Europe | -31 | -46 | -50 | -50 | -33 | -51 | -57 | -58 | -67 | -72 |
| Middle East | 18 | 31 | 51 | 72 | -16 | -19 | -20 | -23 | -37 | -76 |
| North Africa | -9 | 3 | 24 | 53 | -24 | -25 | -16 | -2 | -56 | -71 |
| North America | -28 | -51 | -51 | -51 | -31 | -55 | -59 | -64 | -73 | -78 |
| North Pole | 1 | -1 | -15 | -13 | -15 | -22 | -19 | -23 | -61 | -78 |
| Ocean | -6 | -0.2 | 11 | 25 | -14 | -22 | -29 | -27 | -51 | -64 |
| Pacific Aus NZ | -20 | -31 | -32 | -33 | -28 | -53 | -58 | -63 | -72 | -84 |
| Russia Bel Ukr | -1 | -4 | -9 | -8 | -18 | -28 | -29 | -35 | -62 | -74 |
| Southern Africa | 10 | 13 | 30 | 49 | -12 | -21 | -18 | -12 | -41 | -50 |
| South America | -6 | 1 | 15 | 28 | -9 | -11 | -6 | -2 | -46 | -66 |
| South Asia | 19 | 67 | 139 | 199 | -1 | 12 | 41 | 66 | -29 | -48 |
| South East Asia | 24 | 45 | 71 | 101 | -1 | -7 | -5 | 1 | -35 | -59 |





**Table 5. Monthly and annual mean surface O₃ changes (ppbv) from the TF-HTAP2 multi-model mean and the parameterisation over South Asia due to a 20% reduction in anthropogenic precursor emissions over this region. Multi-model mean values are shown with +/- 1 standard deviation for the available TF-HTAP2 models and the parameterised approach is based on multiple models.**

| TF-HTAP2 South Asia Experiment | Surface O₃ response (ppbv +/- one standard deviation) | | | | |
|---|---|---|---|---|---|
| | January | April | July | October | Annual Mean |
| **TF-HTAP2 Multi-model Model**[NB] | -1.67 ± 0.73 | -1.48 ± 0.29 | -1.22 ± 0.21 | -1.72 ± 0.44 | -1.51 ± 0.35 |
| **Parameterisation mean (multi-models)** | -1.58 ± 0.54 | -1.48 ± 0.39 | -1.09 ± 0.33 | -1.89 ± 0.55 | -1.50 ± 0.29 |

5    [NB] – Models contributing to the multi-model mean are C-IFSv2, CAMchem, CHASER_re1, CHASER_t106, GEOSCHEM-adjoint, HadGEM2-ES, OslsoCTM3.v2.





**Table 6. Annual mean surface O₃ change (ppbv plus one standard deviation) in 2030 and 2050 (relative to 2010) for each RCP scenario derived from the parameterisation in this study and that of Wild et al., (2012).**

| CMIP5 RCP | Global Surface O₃ response from 2010 to 2050 (ppbv) | | | |
| | This Study | | Wild et al., (2012) | |
| | 2030 | 2050 | 2030 | 2050 |
|---|---|---|---|---|
| RCP2.6 | -1.5 +/- 0.1 | -2.4 +/- 0.3 | -1.2 +/- 0.3 | -2.0 +/- 0.5 |
| RCP4.5 | -0.1 +/- 0.1 | -1.1 +/- 0.2 | -0.2 +/- 0.2 | -0.8 +/- 0.4 |
| RCP6.0 | -0.5 +/- 0.1 | -0.7 +/- 0.1 | -0.4 +/- 0.1 | -0.4 +/- 0.2 |
| RCP8.5 | +0.7 +/- 0.2 | +1.0 +/- 0.5 | +1.0 +/- 0.2 | +1.5 +/- 0.5 |



**Table 7. Methane lifetime (τ) and feedback factor in TF-HTAP2 models that provided appropriate data (OH and CH₄ concentrations).**

| Model | TF-HTAP2 Experiment | $\tau_{OH}$[1] | $T_{total}$[2] | F[3] |
|---|---|---|---|---|
| CHASER_re1 | BASE | 7.19 | 6.51 | |
| | CH4INC | 7.62 | 6.86 | 1.46 |
| HadGEM2-ES | BASE | 8.8 | 7.8 | |
| | CH4INC | 9.29 | 8.17 | 1.40 |
| | CH4DEC | 8.43 | 7.51 | 1.37 |
| TF-HTAP1 Mean | | 10.19 +/- 1.72 | 8.84 +/- 1.33 | 1.33 +/- 0.06 |

[1] CH₄ lifetime for loss by tropospheric OH (years) defined as atmospheric burden in each experiment divided by the tropospheric CH₄ loss rate with OH with a tropopause of 150 ppb of O₃ used.
[2] Total atmospheric CH₄ lifetime (years) defined as the reciprocal mean of $\tau_{OH}$ and assuming a lifetime in the stratosphere and soils of 120 years and 160 years respectively (Prather et al., 2001).
[3] The feedback factor is the ratio of the atmospheric response (or perturbation) time to the global atmospheric lifetime. It is defined as $f = 1/(1 - S)$ where S is determined from the BASE and CH₄ perturbation simulations and defined as $S = (\delta \ln(\tau))/(\delta \ln[CH4])$ and CH4 abundances for TF-HTAP2 are 1798 ppbv in BASE, 1562 ppbv in CH4DEC and 2121 ppbv for CH4INC (Prather et al., 2001).



**Table 8. Percentage change in global CH$_4$ abundance and global and regional NO$_X$ emissions relative to 2010 over each TF-HTAP2 world region for the different CMIP6 emission scenarios (SSP1 26, SSP2 60 and SSP3 BASE)**

| TF-HTAP2 Region | Annual total emission change (%) from 2010 | | | | | | | | | | | |
| --- | --- | --- | --- | --- | --- | --- | --- | --- | --- | --- | --- | --- |
| | SSP1 26 | | | | SSP2 60 | | | | SSP3 BASE | | | |
| | 2020 | 2030 | 2040 | 2050 | 2020 | 2030 | 2040 | 2050 | 2020 | 2030 | 2040 | 2050 |
| Global CH$_4$ | 1 | -7 | -16 | -23 | 4 | 6 | 7 | 5 | 8 | 18 | 28 | 37 |
| Global NO$_X$ | -8 | -25 | -35 | -48 | -7 | -9 | -16 | -21 | 10 | 14 | 15 | 16 |
| *Regional NO$_X$ Emissions* | | | | | | | | | | | | |
| Central America, South America | -2 | -22 | -27 | -34 | -10 | -11 | -15 | -24 | 13 | 22 | 30 | 36 |
| Central Asia, Rus Bel Ukr | -14 | -32 | -40 | -49 | 1 | 2 | -5 | -14 | -1 | -5 | -5 | -12 |
| East Asia, South Asia, South East Asia | 4 | -8 | -22 | -35 | -3 | -1 | -12 | -22 | 26 | 45 | 54 | 54 |
| Europe, North America, Pacific Aus NZ | -31 | -62 | -68 | -74 | -31 | -43 | -51 | -57 | -8 | -22 | -30 | -32 |
| Middle East, North Africa, Southern Africa | 4 | -3 | -4 | -2 | 3 | 11 | 13 | 12 | 7 | 14 | 26 | 33 |



**Table 9. Multi-mean parameterised responses in annual mean global ozone burden and ozone radiative forcing in 1980 and for the CMIP5 emission scenarios in 2030, with changes calculated relative to the year 2000 for comparison with values from ACCMIP (+/- 1 standard deviation of multi-model responses).**

| | Ozone Burden (Tg) | | | | Ozone Radiative Forcing relative to | |
| | Total | | Change from 2000 | | 2000 (mW m$^{-2}$) | |
| Year | Parameterisation | ACCMIP* | Parameterisation | ACCMIP* | Parameterisation | ACCMIP* |
|---|---|---|---|---|---|---|
| 1980 | 322 | 322 +/- 22 | -10 | -15 | -42 | -59 +/- 59 |
| 2000 | 332 | 337 +/- 24 | 0 | 0 | 0 | 0 |
| RCP2.6 2030 | 318 | 319 +/- 22 | -14 | -18 | -49 | -39 +/- 52 |
| RCP4.5 2030 | 331 | 344 +/- 26 | -1 | +7 | +4 | +37 +/- 45 |
| RCP6.0 2030 | 327 | 336 +/- 31 | -5 | -1 | -17 | -16 +/- 66 |
| RCP8.5 2030 | 340 | 357 +/- 26 | +8 | +20 | +35 | +83 +/- 61 |

5  * - Mean tropospheric Ozone burden and radiative forcing between future year and 2000s from the ACCMIP multi-model ensemble as presented in Young et al., (2013) and Stevenson et al., (2013)



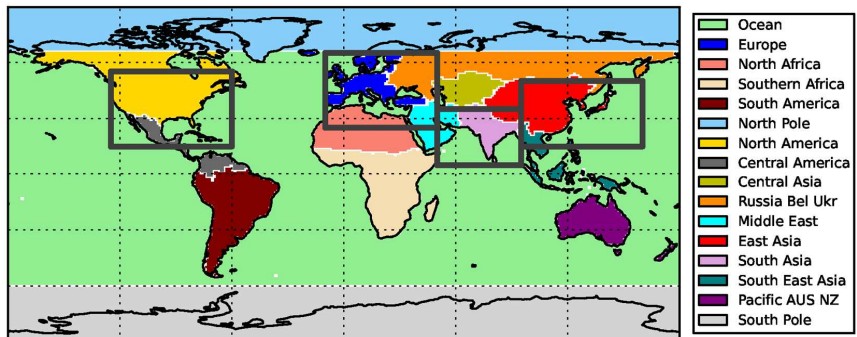

**Figure 1: Source/Receptor regions used in TF-HTAP2 (coloured regions) and TF-HTAP1 (solid grey line boxes) experiments.**





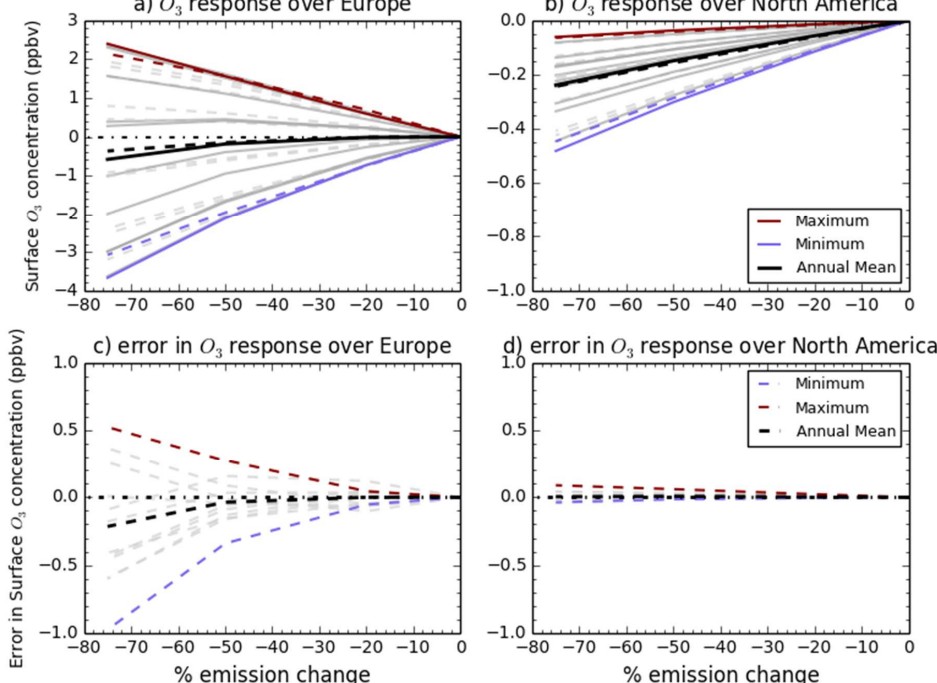

**Figure 2: Sensitivity of monthly surface O₃ changes in HadGEM2-ES (solid lines) and that of the parameterised response using solely HadGEM-ES as input (dashed lines) to 20%, 50% and 75% reduction in all precursor emissions over the European source region (a) and the remote receptor region of North America (b). The difference between HadGEM2-ES and the parameterised response is shown over Europe (c) and North America (d). Annual mean values are in black with monthly responses in grey and the highest and lowest months are highlighted in red and blue.**



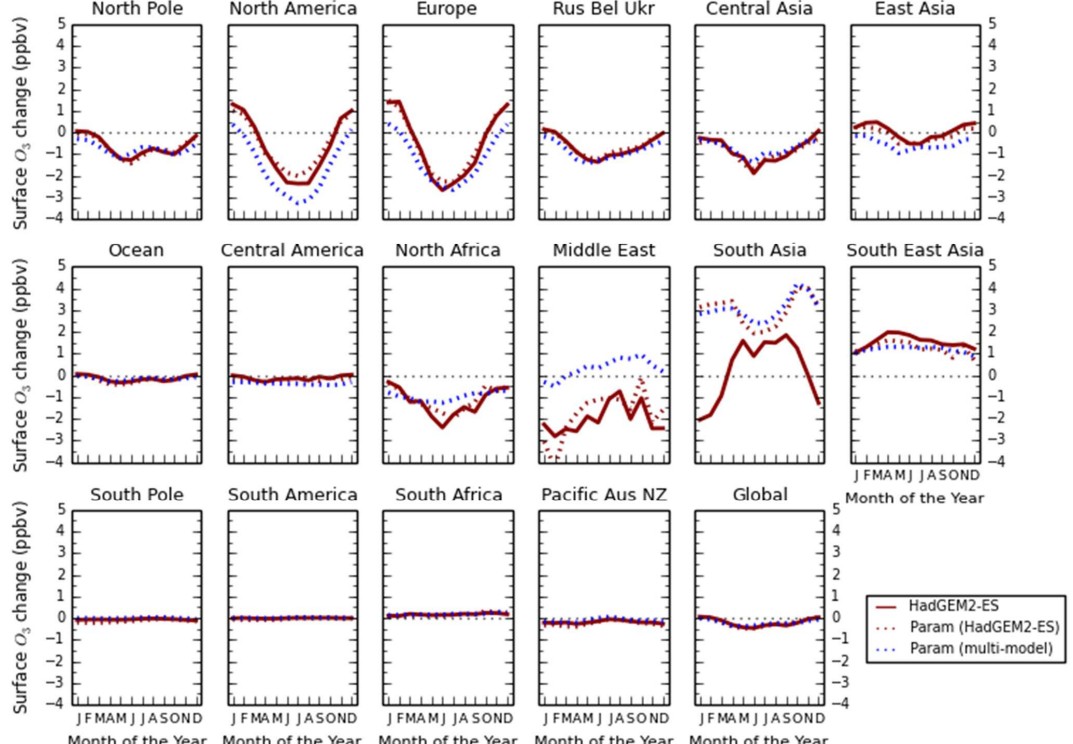

**Figure 3: Monthly mean regional surface O₃ changes between 2010 and 2030 for the ECLIPSE V5a CLE scenario in HadGEM2-ES (red) and the parameterised response based on only HadGEM2-ES inputs (red dashed) and multiple model inputs (blue dashed).**





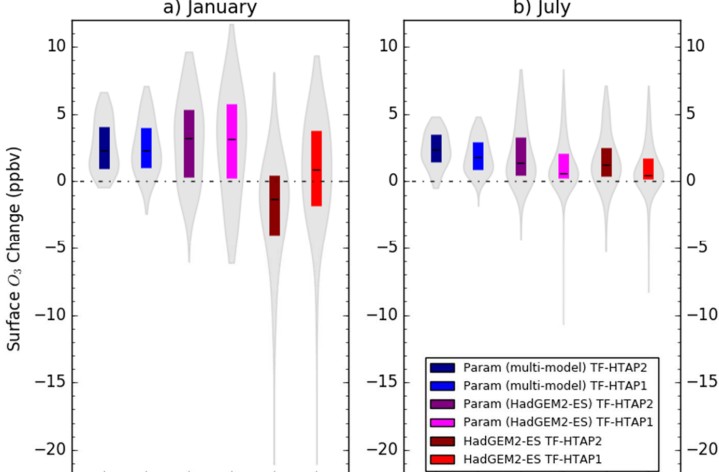

**Figure 4: January (a) and July (b) monthly mean surface O₃ changes over South Asia between 2010 and the ECLIPSE V5a CLE emission scenario in 2030 for HadGEM2-ES and the parameterised response based only on HadGEM2-ES and on multiple models.**
5  **O₃ responses are calculated over the South Asian region as defined in both TF-HTAP1 and TF-HTAP2 (Fig. 1). Grey shading represents the spatial distribution of O₃ changes across all grid boxes, coloured boxes show the range of the 25ᵗʰ to 75ᵗʰ percentile values and the solid line shows the median value over the South Asian region**





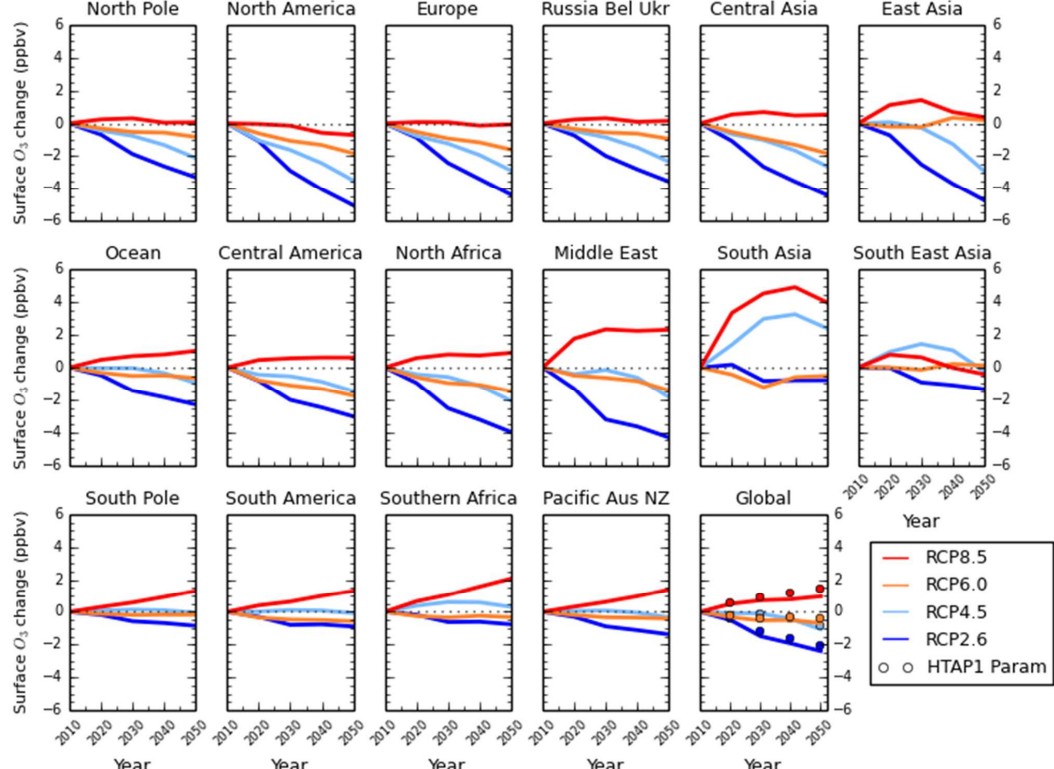

**Figure 5: Annual mean regional surface O₃ changes between 2010 and 2050 from the parameterisation for the CMIP5 emissions scenarios of RCP8.5 (red), RCP6.0 (orange), RCP4.5 (light blue) and RCP2.6 (blue). The global surface O₃ response from the parameterisation of Wild et al., (2012) for each scenario is represented as circles, but due to differences in regional definitions a straightforward comparison with TF-HTAP1 regions (Europe, North America, South Asia and East Asia) is not possible.**





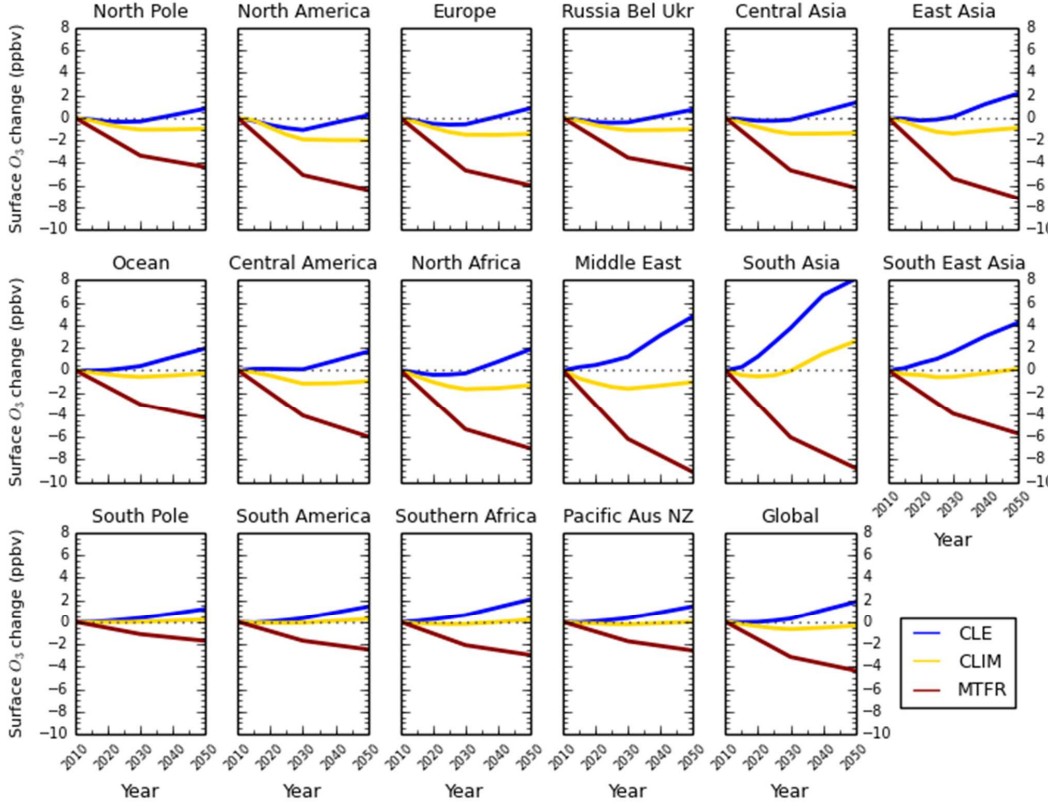

**Figure 6: Annual mean change in regional surface O₃ concentrations between 2010 and 2050 from the parameterisation for the ECLIPSEv5a emissions under the CLE (blue), CLIM (gold) and MTFR (red) scenarios.**





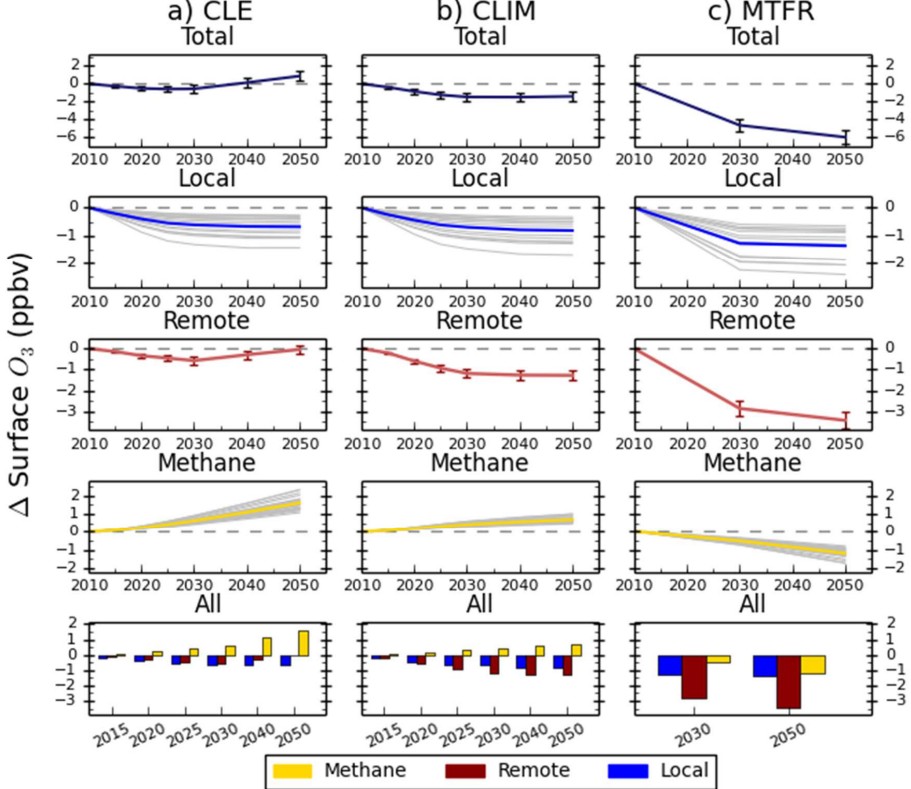

**Figure 7: Total annual mean change in regional surface O₃ concentrations over Europe and the contribution of local (blue), remote (red) and methane (gold) sources between 2010 and 2050 from the parameterisation for the ECLIPSEv5a emissions under the CLE (a), CLIM (b) and MTFR (c) scenarios. Grey lines on the local and methane panels represent individual model estimates of O₃ changes, showing the spread in model responses; Solid lines show the multi-model mean. Error bars represent one standard deviation over the model range. The last row of panels shows the O₃ response from individual sources plotted together for each year.**



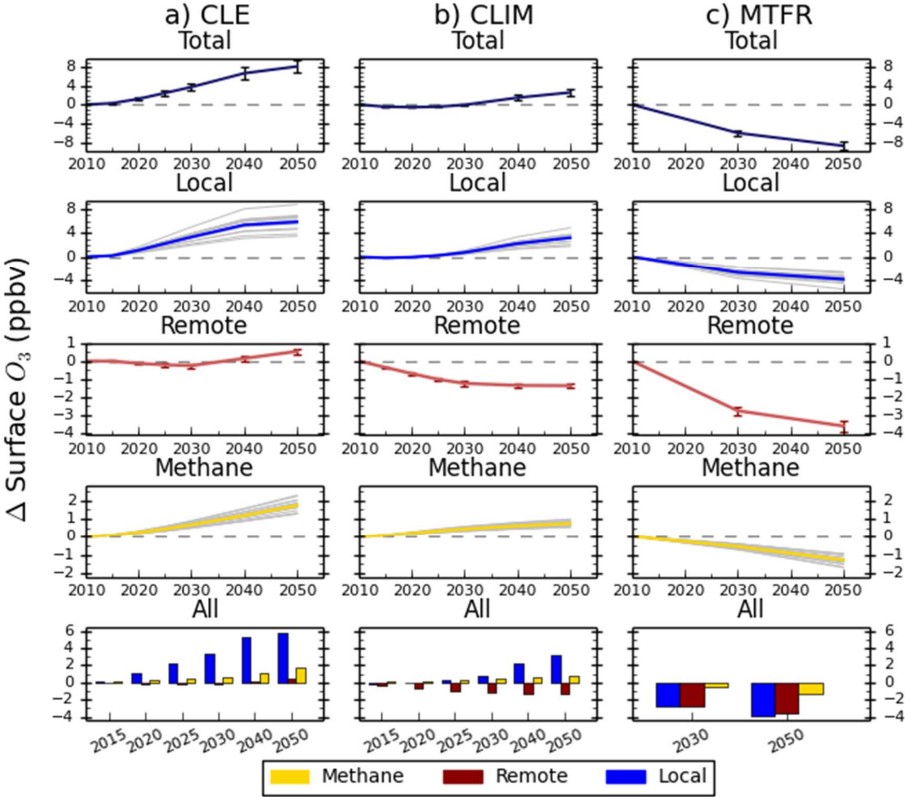

Figure 8: Same as Fig 7. but for the South Asian region.



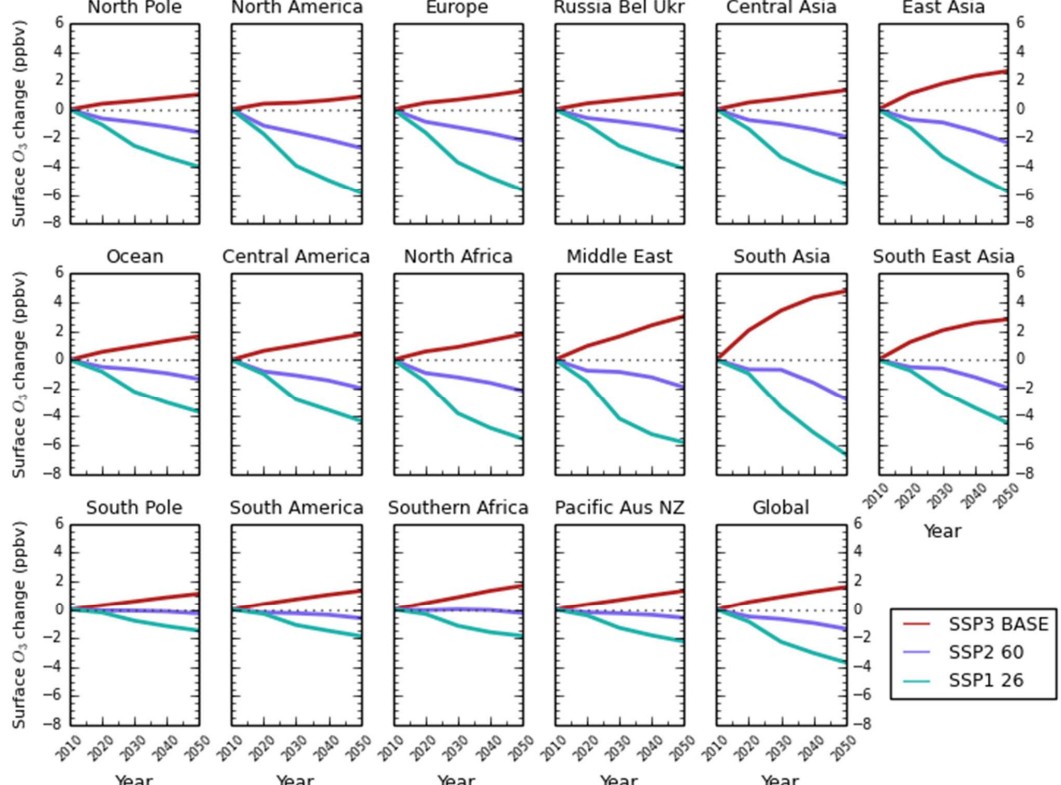

**Figure 9: Annual mean change in regional surface O₃ concentrations between 2010 and 2050 from the parameterisation for the CMIP6 emissions scenarios of SSP3 baseline (red), SSP2 with a radiative forcing target of 6.0 W m⁻² (purple) and SSP1 with a**
5   **radiative forcing target of 2.6 W m⁻² (green).**





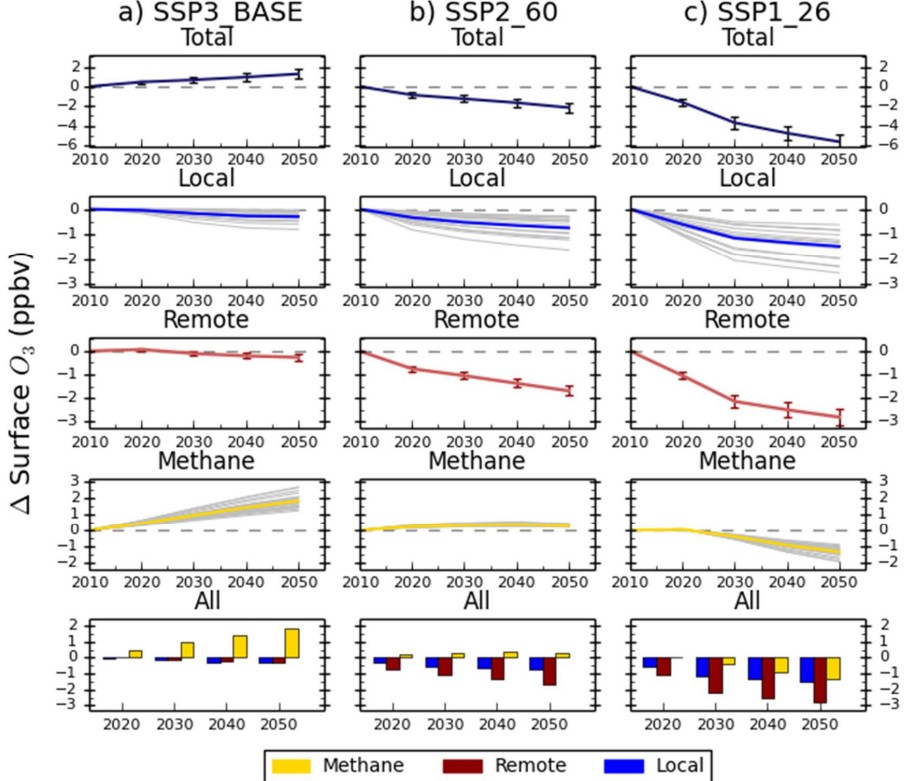

**Figure 10: Total annual mean change in regional surface O₃ concentrations over Europe and the contribution of local (blue), remote (red) or methane (gold) sources between 2010 and 2050 from the parameterisation for the CMIP6 emissions scenarios of SSP3 BASE (a), SSP2 6.0 (b) and SSP1 2.6 (c). Grey lines on the local and methane panels represent individual model estimates of O₃ changes, showing the spread in model responses; solid lines show the multi-model mean. Error bars represent one standard deviation over the model range. The last row of panels shows the O₃ response from individual sources plotted together for each year.**



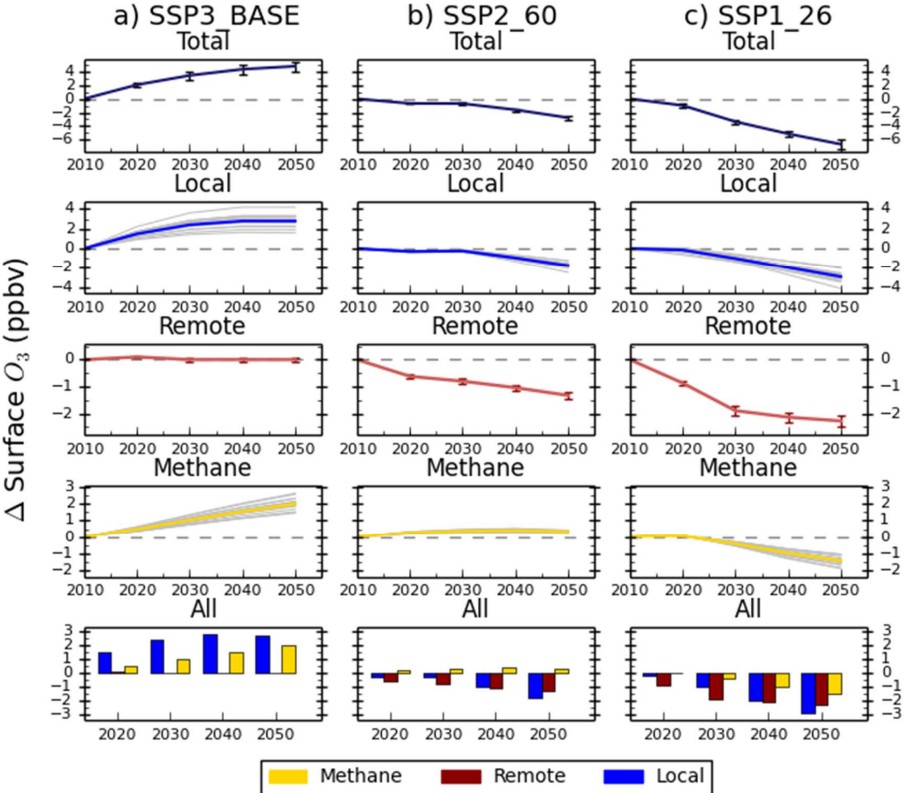

Figure 11: Same as Fig. 10 but for the South Asian region.





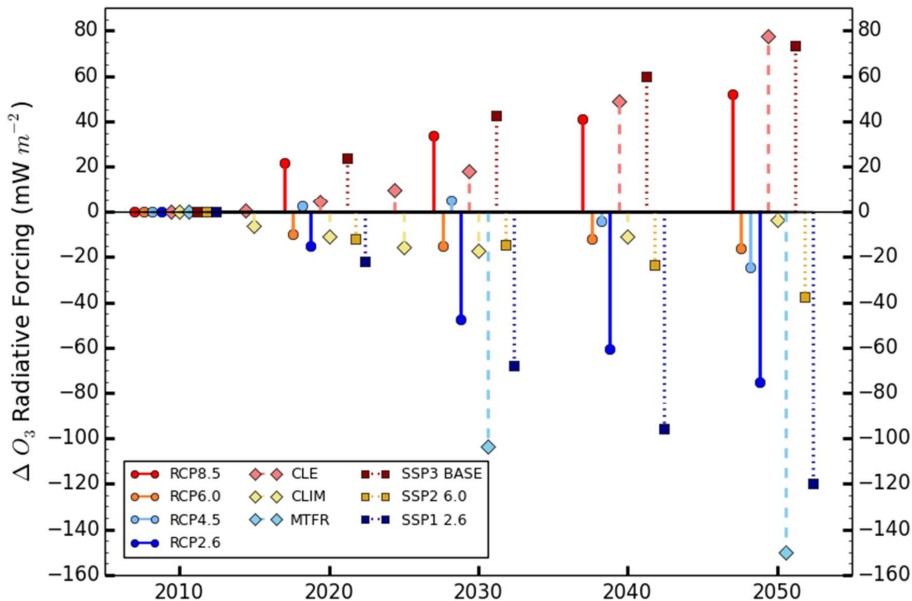

**Figure 12. Parameterised response in the global annual mean ozone radiative forcing relative to 2010 for the different CMIP5 RCPs (circles), ECLIPSE (diamonds) and CMIP6 SSPs (squares).**





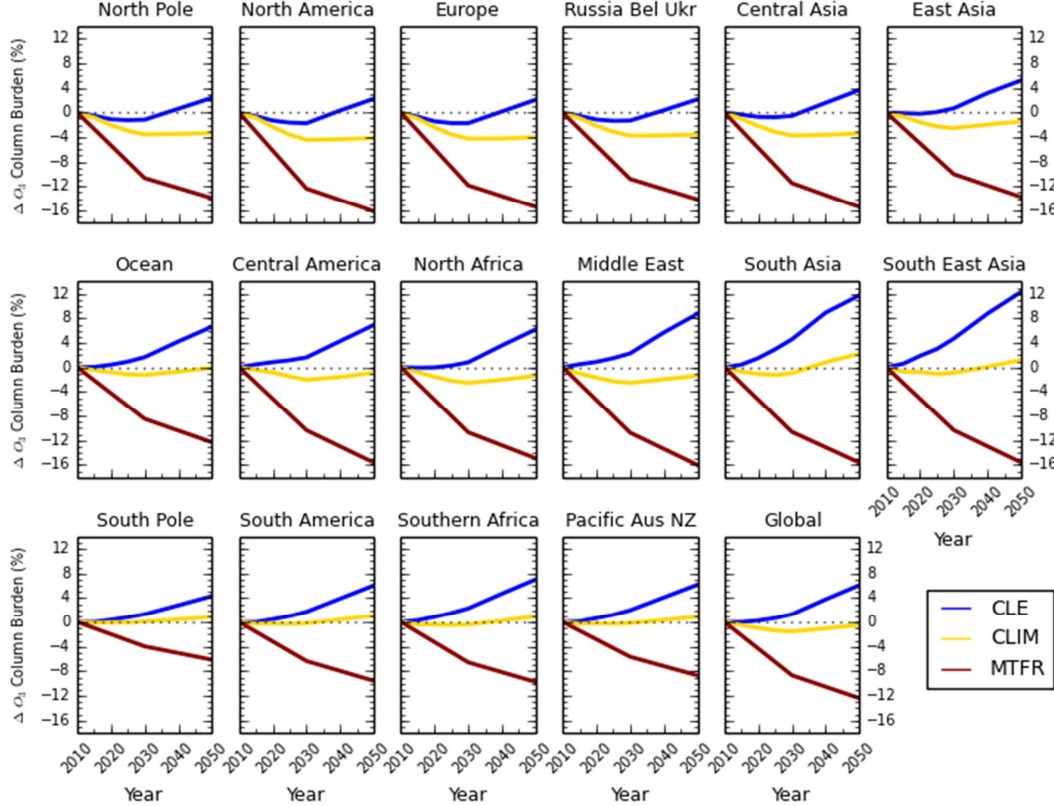

**Figure 13: Annual mean percentage change in the regional and global tropospheric O₃ burden over the period 2010 to 2050 from the parameterisation for the ECLIPSEv5a emissions under CLE (blue), CLIM (gold) and MTFR (red) scenarios.**





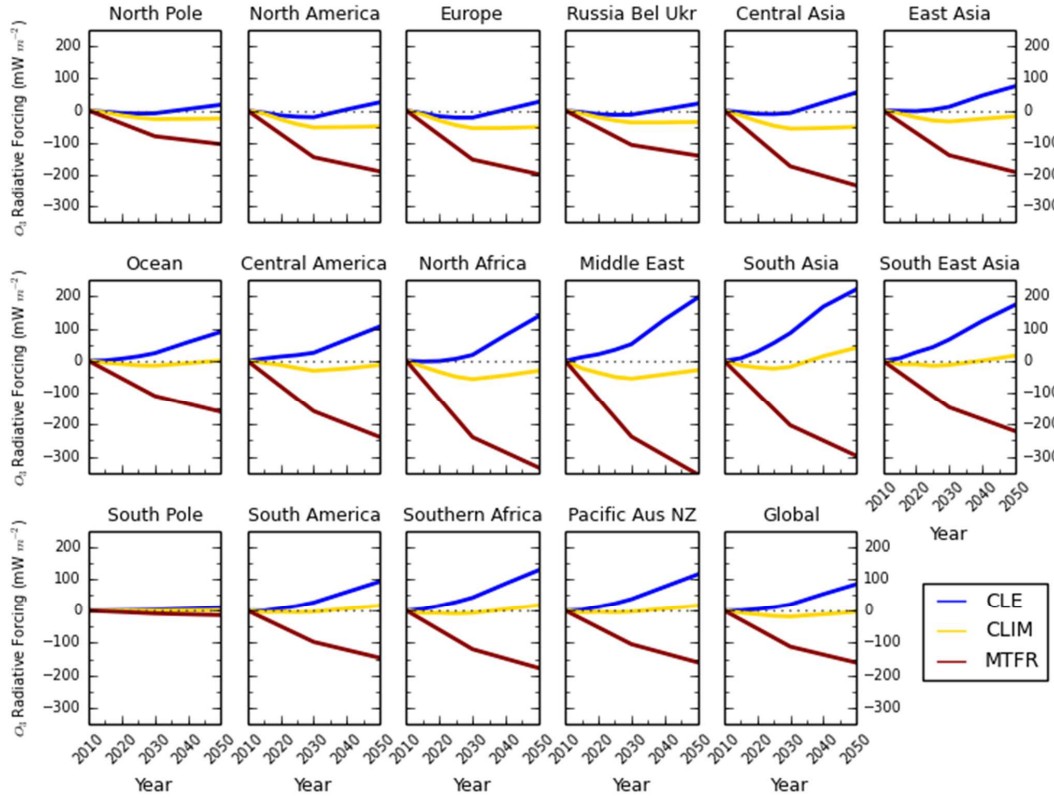

**Figure 14: Annual mean regional O₃ radiative forcing relative to 2010 from the parameterisation for the ECLIPSEv5a emissions under the CLE (blue), CLIM (gold) and MTFR (red) scenarios.**

