# Peer review of "The Impact of Future Emission Policies on Tropospheric Ozone using a Parameterised Approach"

_Atmospheric Chemistry and Physics, 2017_

## Referee Comment (RC1) · Anonymous Referee #1 · 7 Mar 2018

Overall, I think this will be an important and useful paper detailing the impact of a great many future emission scenarios on surface ozone and radiative forcing. In general, the paper is well written with lots of detailed tables and clear figures. I would recommend publication following the changes detailed below.

Major Comments

1. Overall, I found the explanation for the parameterization rather confusing and not straightforward. Please think about how to make the explanation more precise. Some details...P4, L5: "the scale factor, f, is replaced by g". This is a rather confusing way to put it. Why don't you write out the full parameterization from the beginning by defining the various terms in equation (2) dependent on the constituent and not explain the parameterization by first defining f, then replacing f with g? Once could easily expand

equation (2) to include the definitions for the various constituents. In addition, equation (2) is written as one might write out a computer code, but does not make sense from a mathematical viewpoint. Where does the factor (2f-g) come from. The factor g is evidently different for both CH4 and NOx from that given in equation 2? (equation 3). This should be discussed at the beginning and not mid-way through the paper. Where exactly is the ozone adjustment factor used (page 10)? Section 2.1 is titled "Original Ozone Parameterization", but as far as I understand it is also the parameterization used in the present paper.

2. I think the paper could do a better job of emphasizing which results should be believed and which should be treated with skepticism. There are a number of emission scenarios where the emission change is over 50% (either with a positive or a negative change). To what extent should the results from these scenarios be believed? Results that should be treated with caution could be clearly indicated in the tables. The radiative forcing calculation does not seem particularly accurate. The authors claim that the parameterization reproduces the ACCMIP changes fairly well (p13, l23), but if I understand correctly the parameterized radiative forcing should be 20 to 30 MW m-2 larger than the ACCMIP results (as it does not account for the climate feedbacks which represent a large part of the ACCMIP signal). Thus, it looks like in most cases the future mean radiative forcing is dramatically underestimated (although, perhaps with the extremely large error bars in ACCMIP it is difficult to really say anything meaningful). Unless I missed it, the authors compared the change in the ozone burden between the parameterization and HadGEM2-ES but not the overall radiative forcing. Without some more evidence, and in a context that does not assume changes in climate, it is somewhat difficult to see what the parameterized radiative forcing calculation adds. I am willing to be convinced otherwise, but do need some convincing. The paper often makes somewhat vague statements about the comparison of the parameterization to explicit results (e.g., it states that the parameterization is valid, or compares well . . .). It would be nice to see in the conclusion a somewhat more explicit discussion of when and under what scenarios the parameterization should be believed: e.g., should it be

believed under scenarios with large changes (e.g., +/- 50%), over regions where decreasing NOx increases ozone, over southeast Asia even with small emission changes, for the radiative forcing in 2100 given the strong climate influence etc? In other words, the certainty bounds should be discussed and quantified in more detail with an overall summary given in the conclusions.

Minor Comments:

Page 1:

L26 "are valid". This is really rather strong language as the accuracy of the parameterization differs depending on the region. It would be better to quantify this a bit more, saying instead something like "are reasonably accurate for most regions" or "are within the model spread for most regions". Once you say they are valid, it is difficult to quantify how valid are they?

L24. The neglect of climate change is mentioned in regards to radiative forcing but not to changes in surface ozone concentration. It would be important to emphasize that changes in climate and associated changes in climate dependent precursor emissions are neglected at the outset (e.g., in L24).

L32 "across different regions". This is a bit confusing. It might be better to say: "will regionally increase by 1 to 8 ppbv".

L33 I wander if it would be clearer to say "change in radiative forcing. . . from 2010 to 2050"?

L31: which regions are not sources?

Page 5:

L3. "Models covered". Rather awkward English usage. Models don't cover. . .

L15-16. It would be worthwhile to add a line that the paper discusses in detail how the results TF-HTAP2 are incorporated below.

L25. It is not clear how the parameterization has been improved.

L24-34. In Table 1 the emission differences are averaged for both the MACCity and EDGAR inventories? It is not clear why the authors did not average the emission differences as used in the TF-HTAP2 models versus those in the TF-HTAP1 models. It is unclear how or why internal consistency (whatever that means) should be relevant here. The actual difference in emissions would seem to be more relevant in comparing the change in O3.

L35 - how many models contributed to the parameterized ozone response? -In general it is not really clear what was done here. Were the emission differences in table 1 used to compute the parameterized change in ozone in table 2? Was the parameterized response from each model computed separately to give the standard deviation in the parameterized response?

L37-38. "similar to the TF-HTAP2 multi-model mean values". This seems a little misleading as the parameterized responses are also similar to the TF-HTAP1 values. It would be more insightful to quantify the extent to which the parameterization quantifies the changes between TF-HTAP1 and TF-HTAP2.

L5. "adjusted". I assume the authors explain how the emissions are adjusted below.

L11-21. Both the source and receptor regions are changed between HTAP1 and HTAP2. It is unclear from the description here how you discretize the response in the HTAP1 models into both smaller source regions and smaller receptor regions. The discussion in 2.3.2 seems to only concern the source region adjustment. Sections 2.3.2 and 2.3.3 should be clarified as to the exact procedure used.

-L16-23. This seems to be largely a repeat of what is said above.

-L17 "significant improvements". In what way? Are more models are used, or are the source-receptor regions are better defined, or do the authors feel the parameterization itself has been improved in some fundamental way? The improved parameterization is again mentioned on page 9, line 19 (and probably elsewhere). Please be explicit on how exactly the parameterization is improved.

-L31 "is working well". This is a little hard to tell from the figures. It would be valuable to show the percentage error as a function of month for the two responses.

Page 8.

-The results over South Asia are really quite strange as the parameterization based on HadGEM2 fits the multi-model parameterization. The authors seem to be arguing on page 8 and 9 that this is a difficult region to simulate and that perhaps it is not surprising that the parameterization based on HadGEM with the large titration might not be able to simulate this region accurately. However, this seems to be only half of the story.... Why does the parameterization based specifically on HadGEM match the multi-model parameterization? And does the multi-model parameterization capture the multi-model response in this region?

-L25-26: "compare well with ACCMIP multi-model means for intermediate emission scenarios..." What about RCP4.5? Isn't this a intermediate emission scenario? The comparson from RCP4.5 does not look that good.

---

## Referee Comment (RC2) · Anonymous Referee #2 · 21 Mar 2018

The manuscript by Turnock et al. presents an updated version of earlier work done by Wild et al. (2012), who constructed a parameterisation for calculating the response of tropospheric ozone to changes in precursor emissions (and methane abundances). The parameterisation reduces the need to run an ensemble of computationally expensive global models of atmospheric chemistry in order to explore the effects of different emission scenarios on the abundance of tropospheric ozone, and the need to run multiple model experiments to determine the influences of emissions from multiple source regions on individual receptor regions. As such, this is a tool for rapid assessment of alternative emission control policies, but does not replace global atmospheric chemistry models, since it does not represent the nonlinearities of tropospheric ozone chemistry, or the influence of future climate changes on tropospheric ozone. As an update to Wild

et al. (2012), I expect that this paper will be widely used.

Unfortunately I found much of the description of the method to be vague and confusing. This should be improved before the paper is published. Also, the authors could do more to compare the predictions of their parameterised ozone with actual simulations from global atmospheric chemistry models. More details are given below

Page 3, lines 5-7: Ignoring changes in future ozone due to climate change is an important limitation of this study. Here it would be appropriate to give a short summary of the expected changes in surface ozone due to climate change, to provide the reader with more information about this limitation.

Page 3, line 13: There appears to be a typo in the last sentence of this paragraph.

Page 4, line 5: Why is f replaced with g? Is this just a difference in terminology between Wild et al. (2012) and this study? Or something else?

Page 3, line 9: Where does "2f -g" come from? This appears to come out of nowhere.

Page 3, line 10: Much more explanation is needed here. Which model simulations? Which year? Why is the spatial extent of the titration regimes important? How is the magnitude defined? A lot of very important information appears to have simply been left out.

Page 3, line 14: It's fascinating to read that the response of surface ozone to methane, a well-mixed gas with a lifetime measured in years, could be similar to the response to NOx, which has a lifetime on the order of hours. In what way is the response "similar"? More explanation is needed here.

Equation 2: How are the coefficients in the calculation of g determined?

Page 8, line 40: There is no need to mention titration a second time.

Section 3.3.1: Can you compare the predictions of the parameterisation using the RCP scenarios with the actual global model runs done in the ACCMIP exercise? See Young

et al. (2013) for some examples. This comparison would really help the reader to understand more about how well the parameterisation is doing in comparison with the global models. Perhaps this comparison could be added to Fig. 5. Also, why isn't this part of Section 4? ("Future Surface Ozone Predictions")

Page 10, line 8: "successive emission increases" of what magnitude? In what succession?

Equation 3: The coefficients appear to have changed since Equation 2. Why? How are they calculated? This seems like something for Section 2.

Page 10, line 33: What are the "appropriate fields"?

Section 4.1: Can you compare the predictions of the parameterisation using the ECLIPSE scenarios with the actual global model runs done in the ECLIPSE exercise? See Stohl et al. (2015, DOI: 10.5194/acp-15-10529-2015) and work citing that paper for examples.

---

## Short Comment (SC1) · 21 Mar 2018

In general this is a well-written and very useful paper that addresses relevant policy issues.

As a possible user of the ozone precursor source-receptor relations, I would like to make some suggestions that would improve the readability of the paper and create the possibility for the scientific community to replicate the results.

Eq. 1: the same variable symbol (deltaO3) is used at left and right-hand side of the equation, while they have different meanings. The same observation can be made for Eq. 2 where e.g is written fij = 2fij - gij; suggest to use a different symbol at the left hand side.

[Figure]

Eq. 1 expresses deltaO3 as response to the sum of an emission change (for NOx, CO and NMVOCs), and an abundance change in CH4. For the user, using emission changes for all precursors would make more sense. Isn't it possible, from the box model mentioned in section 3.2, and using a feedback factor, to relate a change in abundance to a change in emissions? Why not normalise the source-receptor responses by the emission strength? It would be useful to emphasise the time scale of the CH4 responses and how to deal with this in such a parametrised approach.

It's not clear why paragraph 3.1 is named 'Scaling Factors'

Page 10, line 12: 'the same scaling factor', is not clear if 'same' refers to using the same as in HTAP1, or using the same (new) factor for CH4 and NOx. So, Eq. 3: is this now the scaling factor replacing the 0.95f+0.05fˆ2 from HTAP1 both for NOx and CH4?

Figures 7 and 8 (and similar in SI): does the ozone trend from CH4 include the transient effect of the 12y perturbation response time? How can Eq. 3 be applied (for CH4) to obtain this trend? The figures show the change in ozone relative to year 2010; does it include the time-lagged impact of CH4 emissions before that date? I would appreciate having the box model for CH4 better documented.

---

## Author Comment (AC1) · 3 May 2018

**Author's response to referee comments on "The Impact of Future Emission Policies on Tropospheric Ozone using a Parameterised Approach"**

S. T. Turnock et al.
Correspondence to: S. T. Turnock
(steven.turnock@metoffice.gov.uk)

We would like to thank all of the reviewers for their helpful and constructive comments. Below we have responded to each comment in turn and made alterations to the manuscript where appropriate (shown enclosed in "*speech marks and italic font*" *and any deletions from the manuscript shown with a strikethrough ""*). The referee comments are shown first in grey shading and the author's response is shown below in normal font.

**Response to Referee 1**

Overall, I think this will be an important and useful paper detailing the impact of a great many future emission scenarios on surface ozone and radiative forcing. In general, the paper is well written with lots of detailed tables and clear figures. I would recommend publication following the changes detailed below.

Major Comments

1. Overall, I found the explanation for the parameterization rather confusing and not straightforward. Please think about how to make the explanation more precise. Some details…P4, L5: "the scale factor, f, is replaced by g". This is a rather confusing way to put it. Why don't you write out the full parameterization from the beginning by defining the various terms in equation (2) dependent on the constituent and not explain the parameterization by first defining f, then replacing f with g? Once could easily expand equation (2) to include the definitions for the various constituents. In addition, equation (2) is written as one might write out a computer code, but does not make sense from a mathematical viewpoint. Where does the factor (2f-g) come from. The factor g is evidently different for both CH4 and NOx from that given in equation 2? (equation 3). This should be discussed at the beginning and not mid-way through the paper. Where exactly is the ozone adjustment factor used (page 10)? Section 2.1 is titled "Original Ozone Parameterization", but as far as I understand it is also the parameterization used in the present paper.

We would like to thank the reviewer for their useful comments on the initial description of the parameterisation and appreciate some of the confusion in the description. The parameterisation used in this study is fully consistent with that of Wild et al., (2012) but we have updated the notation to make it easier to follow.

Section 2.1. has been renamed as "*Parameterisation of Ozone*".

To avoid some of the confusion with equation 2 in the manuscript a new fractional emission change factor (r) is defined as below and becomes equation (1):

$$r_{ij} = \frac{\Delta E_{ij}}{-0.2 * E_{ij}}$$  (1)

Equation (2) in the original manuscript has been replaced with the following three equations (which now become equations 3, 4 and 5) to clearly identify the different emission scaling factors and when they are used in the parameterisation.

$$f_{ij} = r_{ij} \qquad \text{\textit{Linear Scaling of } } O_3 \text{\textit{ response}}$$ (3)

$$f_{ij} = 0.95r_{ij} + 0.05r_{ij}^2 \quad \text{\textit{Scaling accounting for reduced } } O_3 \text{\textit{ increases from }} NO_X \text{\textit{ and }} CH_4$$ (4)

$$f_{ij} = 1.05r_{ij} - 0.05r_{ij}^2 \quad \text{\textit{Scaling for titration regimes where decreasing }} NO_X \text{\textit{ increases }} O_3$$ (5)

In addition, a schematic (shown below) has now been included in the supplementary as Figure S1 to show the various steps to the parameterisation and when to use the different scaling factors.

**1. Calculate fractional emission change (r)**

$$r_{ij} = \frac{\Delta E_{ij}}{-0.2 \times E_{ij}}$$

**2. Generate scaling factors (f)**

**2.1. Linear scaling of $O_3$ response**

$$f_{ij} = r_{ij}$$

**2.2. Non-linear scaling accounting for reduced $O_3$ increases from NOx and CH$_4$**

$$f_{ij} = 0.95r_{ij} + 0.05r_{ij}^2$$

**2.3. Non-linear scaling for titration regimes where decreasing NOx increases $O_3$**

$$f_{ij} = 1.05r_{ij} - 0.05r_{ij}^2$$

**3. Apply different scaling factors to precursors and source regions under different chemical regimes**

[Figure]

**$O_3$ Production Regime**

Precursor Perturbation:

CO and NMVOCs – Linear scaling (1)

NOx and CH$_4$ – Non-linear scaling (2)

**$O_3$ Titration Regime**

Precursor Perturbation:

Increased NOx – Linear scaling (1)

Decreased NOx – Non-linear scaling (3)

**Figure S1** – Schematic showing the different steps used in the parameterisation from calculating a fractional emission change (1), to generating an emission scaling factor (2) and applying this to the appropriate precursor in a particular chemical regime (3). The figures at the bottom illustrate the effect of applying the quadratic function compared to the linear one in the different chemical regimes.

We have amended Section 2.1 to make it clearer and easier to read and reflect the changes above. For the convenience of the reviewer we present the entire revised section 2.1 below.

**"2.1 Parameterisation of Ozone**

*The parameterisation developed in this study is based on an earlier version developed from the TF-HTAP1 experiments by Wild et al., (2012). This simple parameterisation enabled the regional response in surface $O_3$ concentrations to be estimated based on changes in precursor emissions and $CH_4$ abundance. The input for this parameterisation came from 14 different models that contributed to TF-HTAP1. All the models ran the same emission perturbation experiments (20% reduction in emissions of oxides of nitrogen ($NO_X$), carbon monoxide (CO), non-methane volatile organic compounds (NMVOCs) individually and all together) over the four major northern hemisphere source regions of Europe, North America, East Asia and South Asia. Additional experiments included global perturbations of emission precursors (E), as well as a 20% reduction in global $CH_4$ abundance. The multi-model $O_3$ responses from the 20% emission perturbation experiments ($\Delta O_{3e}$ for emissions of NOx, CO and NMVOCs and $\Delta O_{3m}$ for $CH_4$) are then scaled by the fractional emission changes (r) from a given emission scenario over each source region (Eq. 1).*

$$r_{ij} = \frac{\Delta E_{ij}}{-0.2 * E_{ij}} \tag{1}$$

*The monthly mean $O_3$ response ($\Delta O_3$) is calculated as the sum over each receptor region (k) of the scaled $O_3$ response from each model to the individual precursor species (i - CO, $NO_X$ and NMVOCs) in each of the five source regions (j - Europe, North America, East Asia, South Asia and rest of the world), including the response from the change in global $CH_4$ abundance (Eq. 2, reproduced from Wild et al., 2012).*

$$\Delta O_3(k) = \sum_{i=1}^{3}\sum_{j=1}^{5} f_{ij}\Delta O_{3e}(i,j,k) + f_m \Delta O_{3m}(k) \tag{2}$$

$$f_{ij} = r_{ij} \qquad \textit{Linear Scaling of } O_3 \textit{response} \tag{3}$$

$$f_{ij} = 0.95 r_{ij} + 0.05 r_{ij}^2 \quad \textit{Scaling accounting for reduced } O_3 \textit{ increases from } NO_X \textit{ and } CH_4 \tag{4}$$

$$f_{ij} = 1.05 r_{ij} - 0.05 r_{ij}^2 \quad \textit{Scaling for titration regimes where decreasing } NO_X \textit{ increases } O_3 \tag{5}$$

*A linear emission scale factor (Eq. 3) is used in Eq. 2 for each emission scenario involving the precursor emissions CO and NMVOCs and is defined as the ratio of the fractional emission change to the 20% emission reduction in the TF-HTAP1 simulations. A similar scale factor for methane ($f_m$) is based on the ratio of the change in the global abundance of $CH_4$ to that from the 20% reduced $CH_4$ simulation ($\Delta[CH_4]/-0.2 \times [CH_4]$). Perturbations to emissions of CO, $NO_x$ and NMVOCs induce a long-term (decadal) change in tropospheric $O_3$ from the change in the oxidising capacity of the atmosphere (OH) and the $CH_4$ lifetime (Wild and Akimoto, 2001; Collins et al., 2002; Stevenson et al., 2004). The long-term impacts from 20% global emission reductions can reduce the $O_3$ response by 6-14% from $NO_x$ emission changes and increase the $O_3$ response by 16-21% from CO changes (West et al., 2007). This long-term response is not accounted for in the simulations used here as $CH_4$ abundances are fixed.*

*Wild et al., (2012) found that this simple linear scaling relationship between emissions and surface $O_3$ was sufficient for small emissions perturbations, but that the relationship started to exhibit larger non-linear behaviour for larger perturbations, particularly for $NO_x$. The linear scaling factor was found to be sufficient for the surface $O_3$ response from emission perturbations of CO and NMVOCs as non-linear behaviour from these precursors is small (Wu et al., (2009). To account for non-linear behaviour of*

*surface $O_3$ to $NO_x$ emission changes, a quadratic scaling factor (Eq. 4) is used, based on additional simulations of surface $O_3$ response over a larger range of emission perturbations in Wild et al., (2012).*

*For the special case of source regions that are under titration regimes, where a reduction in $NO_x$ emissions may lead to an increase in $O_3$, the curvature of the response is reversed for NOx emission decreases (Eq. 5), as described in Wild et al., (2012). A linear scale factor is used for emission increases under these conditions (Eq. 3). The spatial extent of ozone titration is assumed constant as the parameterisation is based on differences between two model simulations and is therefore unable to represent any future changes in chemical regime.*

*The surface $O_3$ response to changes in global $CH_4$ abundances shows a similar degree of non-linearity as that from changes in $NO_x$ emissions (Wild et al., 2012). Therefore, the non-linear scale factor (Eq. 4) is also used to represent the $O_3$ response to changes in $CH_4$ abundances.*

*In summary, the surface $O_3$ response to CO and NMVOC emission perturbations is represented by the linear scale factor (Eq. 3) and to changes in NOx emissions and $CH_4$ abundances by the non-linear scale factor (Eq. 4). For source regions under titration regimes, the surface $O_3$ response to $NO_x$ emissions is limited by Eq. 5 for emission increases but uses the linear scale factor (Eq. 3) for emission increases. The parameterisation is represented schematically in Figure S1 of the supplementary material."*

Equation 3 was generated from TF-HTAP2 models to test whether the non-linear representation from Wild et al., (2012) was still valid. Since the response in this equation is comparable to that in Wild et al., (2012) it was decided for consistency to retain the same non-linear scaling factor and as such is presented in Section 3.3 as a comparison to TF-HTAP1. The adjustment factor on page 10 was only used to compare the $O_3$ response from $CH_4$ between TF-HTAP1 and TF-HTAP2 models. The incorporation of $O_3$ response to $CH_4$ within the parameterisation are described within Section 2.3.3 (Additions from TF-HTAP2) on Page 7 Lines 1 to 4.

2. I think the paper could do a better job of emphasizing which results should be believed and which should be treated with skepticism. There are a number of emission scenarios where the emission change is over 50% (either with a positive or a negative change). To what extent should the results from these scenarios be believed? Results that should be treated with caution could be clearly indicated in the tables.

The radiative forcing calculation does not seem particularly accurate. The authors claim that the parameterization reproduces the ACCMIP changes fairly well (p13, l23), but if I understand correctly the parameterized radiative forcing should be 20 to 30 MW m-2 larger than the ACCMIP results (as it does not account for the climate feedbacks which represent a large part of the ACCMIP signal). Thus, it looks like in most cases the future mean radiative forcing is dramatically underestimated (although, perhaps with the extremely large error bars in ACCMIP it is difficult to really say anything meaningful). Unless I missed it, the authors compared the change in the ozone burden between the parameterization and HadGEM2-ES but not the overall radiative forcing. Without some more evidence, and in a context that does not assume changes in climate, it is somewhat difficult to see what the parameterized radiative forcing calculation adds. I am willing to be convinced otherwise, but do need some convincing.

The paper often makes somewhat vague statements about the comparison of the parameterization to explicit results (e.g., it states that the parameterization is valid, or compares well : : :). It would be nice to see in the conclusion a somewhat more explicit discussion of when and under what scenarios the parameterization should be believed: e.g., should it be believed under scenarios with large changes (e.g., +/- 50%), over regions where decreasing NOx increases ozone, over southeast Asia even with small emission changes, for the radiative forcing in 2100 given the strong climate influence etc? In other words, the certainty bounds should be discussed and quantified in more detail with an overall summary given in the conclusions.

In section 3.1 we evaluate the parameterisation for emission reductions of 20%, 50% and 75% over Europe, a region where deviation from linear behaviour can be large. Detailed testing and evaluation of the limitations of the parameterisation was carried out in Wild et al., (2012), where emission perturbations ranging from a doubling of emissions to a complete removal were undertaken. Wild et al., (2012) found that errors remained below 1 ppbv for emission changes of less than +/- 60% and we find very similar results, with regional monthly mean errors below this level even with a 75% emission reduction (Fig. 2). We do not expect the parameterisation to work as well for large emission perturbations of greater than 75% or for source regions under a titration regime. The limitations of the parameterisation are further discussed in Section 3.2 and summarised on P9 Lines 13 to 16 where the parameterisation is compared and evaluated against global model simulations. But based on the reviewers comments we have attempted to make the limitations of the parameterisation clearer in the manuscript by including the following changes.

P7 line 30 has been amended to:

*"This small internal error between the parameterisation based on HadGEM2-ES and HadGEM2-ES simulations indicates that the parameterisation of $O_3$ is working well for emission changes at least as great as 50%. This is similar to the results of Wild et al., (2012) where more detailed testing found that that the parameterisation resulted in errors of < 1 ppbv for emission perturbations of up to +/- 60%. Here, monthly mean errors are < 1 ppbv for a 75% emission reduction (Fig. 2). The parameterisation is not expected to perform as well for emission perturbations of larger than +/- 60% and in source regions under titration regimes."*

Discussion of the limitations of the parameterisation have also been added to the conclusion section as described below:

P14 Lines 19 to 20 have been removed

*""*

New text has been inserted at P14 Line 18 of the conclusions to provide a discussion of the limitations:

*"However, larger errors are may occur when using emission changes of greater than +/- 60% and when considering long term future scenarios where there may be a significant influence from climate change. In addition, the parameterisation may not perform well over regions where chemical titration is expected to become dominant in the future under large emission increases e.g. South Asia, as it is based on the ozone responses in 2010."*

Thank you to the reviewer for their useful comments on the comparison to the ACCMIP models. We have completely revisited this section in the manuscript. Results in the APCD manuscript for the comparison to ACCMIP were based on an adjustment to $O_3$ responses from emissions perturbations over the period 2010 to 2030 to correct them to the baseline year of 2000. To provide a more direct comparison with the ACCMIP models and the results from Wild et al., (2012) we have re-calculated the responses using the baseline year from the TF-HTAP1 models (2001) and the fractional changes in total emissions based on the period 2000 to 2030.

The results from this more consistent comparison are presented below as an amendment to Table 6, Table 9 and Figure 5 in the original manuscript. The global annual mean surface $O_3$ response for the CMIP5 scenarios in 2030 and 2050 now shows a very close agreement between the parameterisation in this study and that of Wild et al., (2012). Similarly, the prediction of the change in global surface $O_3$ response by the parameterisation is within the spread of the ACCMIP multi-model mean response for the CMIP5 RCPs in 2030. The calculated change in global $O_3$ burden and $O_3$ radiative forcing for each of the CMIP5 RCPs is now within 1 standard deviation of the ACCMIP multi-model mean response. Differences between the parameterisation and the ACCMIP models are expected as the latter contain the influence of climate change and stratospheric sources on tropospheric ozone. The climate change signal for the ACCMIP models reported in Stevenson et al., (2013) was -24 +/- 27 mW m$^{-2}$ over the period 1850 to 2000 and -25 +/- 25 mW m$^{-2}$ for RCP8.5 for the period 1850 to 2030. Therefore, as expected, there is only a relatively small influence of climate change over the period of 2000 to 2030s.

An additional test was performed for emission perturbations under the ECLIPSEv5a current legislation scenario in 2030 comparing the parameterisation based only on HadGEM2-ES input with HadGEM2-ES simulations. The results (new Table 5 below) show that the parameterisation is able to reproduce the changes seen in the HadGEM2-ES model simulations based only on emission perturbations.

The following amendments have been made to the manuscript figures, tables and text to reflect the above results.

Table 6 has been replaced in the manuscript by the following, now identified as Table 7:

**Table 7. Annual mean surface $O_3$ change (ppbv plus one standard deviation) in 2030 and 2050 (relative to 2000) for each RCP scenario derived from the parameterisation in this study and that of Wild et al., (2012).**

| CMIP5 RCP | Global Surface $O_3$ response from 2000 to 2050 (ppbv) | | | |
| | This Study | | Wild et al., (2012) | |
| | 2030 | 2050 | 2030 | 2050 |
| --- | --- | --- | --- | --- |
| RCP2.6 | -1.1 +/- 0.1 | -1.9 +/- 0.3 | -1.1 +/- 0.3 | -2.0 +/- 0.5 |
| RCP4.5 | -0.1 +/- 0.1 | -0.8 +/- 0.2 | -0.2 +/- 0.2 | -0.8 +/- 0.4 |
| RCP6.0 | -0.4 +/- 0.1 | -0.4 +/- 0.1 | -0.4 +/- 0.1 | -0.4 +/- 0.2 |
| RCP8.5 | +1.0 +/- 0.2 | +1.5 +/- 0.5 | +1.0 +/- 0.2 | +1.5 +/- 0.5 |

Table 9 has been replaced in the manuscript by the following, now identified as Table 10:

**Table 10. Multi-mean parameterised responses in annual mean surface ozone, global ozone burden and ozone radiative forcing in 1980 and for the CMIP5 emission scenarios in 2030, with changes calculated relative to the year 2000 for comparison with values from ACCMIP (+/- 1 standard deviation of multi-model responses).**

| Year | Surface Ozone (ppbv) | | Ozone Burden (Tg) | | Ozone Radiative Forcing (mW m$^{-2}$) | |
|---|---|---|---|---|---|---|
| | Param | ACCMIP* | Param | ACCMIP* | Param | ACCMIP* |
| 1980 | -1.3 | -1.3 +/- 0.4 | -17 | -15 +/- 6 | -67 | -59 +/- 21 |
| 2000 | 0 | 0 | 0 | 0 | 0 | 0 |
| RCP2.6 2030 | -1.1 | -1.5 +/- 0.6 | -12 | -12 +/- 8 | -45 | -45 +/- 30 |
| RCP4.5 2030 | -0.1 | +0.2 +/- 0.5 | +4 | +7 +/- 5 | +14 | +24 +/- 19 |
| RCP6.0 2030 | -0.4 | -0.8 +/- 1.0 | +0.3 | -2 +/- 11 | +2 | -13 +/- 39 |
| RCP8.5 2030 | +1.0 | +1.5 +/- 0.7 | +20 | +23 +/- 7 | +80 | +81 +/- 26 |

* - Mean change in the tropospheric Ozone burden and radiative forcing between 2030 and 2000s from the ACCMIP models that provided results for each year of each scenario, as presented in Table 5 of Young et al., (2013) and Table 12 of Stevenson et al., (2013).

The numbers from Wild et al., (2012) on Figure 5 have been slightly amended as shown below.

[Figure]

**Figure 5**: Annual mean regional surface O$_3$ changes between 2010 and 2050 from the parameterisation for the CMIP5 emissions scenarios of RCP8.5 (red), RCP6.0 (orange), RCP4.5 (light blue) and RCP2.6 (blue). The global surface O$_3$ response from the parameterisation of Wild et al., (2012) for each scenario is represented as circles, but due to differences in regional definitions a straightforward comparison with TF-HTAP1 regions (Europe, North America, South Asia and East Asia) is not possible.

The following new table (Table 5) has been included in the manuscript now to show the comparison of the parameterisation against HadGEM2-ES simulations for the ECLIPSE CLE scenario in 2030, neglecting any influence from climate change occurs.

**Table 5. Parameterised responses based only on HadGEM2-ES input for annual mean surface ozone, global ozone burden and ozone radiative forcing using the ECLIPSE CLE emission scenarios in 2030, with changes calculated relative to the year 2010.**

| Scenario | Surface Ozone (ppbv) | | Ozone Burden (Tg) | | Ozone Radiative Forcing (mW m$^{-2}$) | |
|---|---|---|---|---|---|---|
| | Param[1] | HadGEM2-ES | Param[1] | HadGEM2-ES | Param[1] | HadGEM2-ES* |
| ECL 2030 | -0.21 | -0.20 | -0.93 | -0.95 | -0.6 | -0.9 |

[1] Parameterisation based only on HadGEM2-ES input

* Ozone radiative forcing is calculated by applying the same methodology as in the parameterisation (using the relationship between radiative forcing and tropospheric column O$_3$ change based on multi-model ensemble mean results from ACCMIP

P8, Line 20 to 22 the sentence is amended to:

*"Monthly (Fig. 3) and annual (Table 5) surface O$_3$ changes between 2010 and 2030 over the TF-HTAP2 regions for the ECLIPSE V5a CLE scenario from the HadGEM2-ES simulation are compared to that from the parameterisation (based solely on HadGEM2-ES model responses and based on responses from all models)."*

P13, Line 19 to 31 has been amended to:

*"The change in tropospheric O$_3$ burden and O$_3$ radiative forcing for the ECLIPSE CLE scenario in 2030 from the parameterisation was evaluated against the change from the equivalent HadGEM2-ES simulation (Table 5), a self-consistent test based only on emission perturbations with no influence from climate change. The parameterisation is able to reproduce the change in global tropospheric O$_3$ burden (-0.93 Tg) and O$_3$ radiative forcing (-0.6 mW m$^{-2}$) simulated by HadGEM2-ES (-0.95 Tg and -0.9 mW m$^{-2}$), with any slight differences due to the discrepancies identified over South Asia (Figure 3)."*

P13 Lines 22 to 30 have been amended to:

*"In comparison to the ACCMIP multi-model mean, the predicted changes between 2000 to 2030 in both global annual mean surface O$_3$ and global O$_3$ burden from the parameterisation are within the range of the ACCMIP multi model responses (+/- 1 standard deviation) for all the CMIP5 RCPs (Table 10). The predictions of O$_3$ radiative forcing in 2030 from the parameterisation across all the RCPs, when the influence of climate change is anticipated to be small, are also consistent with those from ACCMIP. The sign and magnitude of change in global O$_3$ burden and O$_3$ radiative forcing with the parameterisation for RCP6.0 is different from the ACCMIP results but is still within the range of model responses, which is the largest for this scenario. The comparison with ACCMIP results shows that the parameterisation is able to reproduce changes in global O$_3$ burden and O$_3$ radiative forcing on near-term timescales, when the influence of climate change is small."*

P14 Lines 24 to 27 have been amended as follows:

*"Tropospheric O$_3$ burden and O$_3$ radiative forcing calculated using the parameterisation are within the spread of the response from the ACCMIP models for all of the CMIP5 RCPs in 2030, where the influence from climate change is anticipated to be small."*

Minor Comments:

Page 1:

L26 "are valid". This is really rather strong language as the accuracy of the parameterization differs depending on the region. It would be better to quantify this a bit more, saying instead something like "are reasonably accurate for most regions" or "are within the model spread for most regions". Once you say they are valid, it is difficult to quantify how valid are they?

Line 26 has been changed to the following:

"*Tests against model simulations using HadGEM2-ES confirm that the approaches used within the parameterisation perform well for most regions*"

L24. The neglect of climate change is mentioned in regards to radiative forcing but not to changes in surface ozone concentration. It would be important to emphasize that changes in climate and associated changes in climate dependent precursor emissions are neglected at the outset (e.g., in L24).

The following sentence has been amended at line 23 to 24:

"*Surface and tropospheric $O_3$ changes are calculated globally and across 16 regions from perturbations in precursor emissions ($NO_x$, CO, VOCs) and methane ($CH_4$) abundance only, neglecting any impact from climate change.*"

L32 "across different regions". This is a bit confusing. It might be better to say: "will regionally increase by 1 to 8 ppbv".

Line 32 has been changed to the following:

"*Emission changes for the future ECLIPSE scenarios and a subset of preliminary Shared Socio-economic Pathways (SSPs) indicate that surface $O_3$ concentrations will increase regionally by 1 to 8 ppbv in 2050*"

L33 I wander if it would be clearer to say "change in radiative forcing from 2010 to 2050"?

The sentence on Line 33 has been amended to:

"*A change in the global tropospheric $O_3$ radiative forcing of +0.07 W m$^{-2}$ from 2010 to 2050 …*"

L31: which regions are not sources?

The Polar regions are not considered as sources as they were not included in the TF-HTAP2 experiments. Line 31 has been altered to include mention of these regions:

"*The source regions were updated in TF-HTAP2 to represent 14 new regions (excluding the North and South Poles), aligned on geo-political and land/sea boundaries (Figure 1).*"

Page 5:

L3. "Models covered". Rather awkward English usage. Models don't cover …

The word covered on Line 3 has been replaced with "*conducted*".

L15-16. It would be worthwhile to add a line that the paper discusses in detail how the results TF-HTAP2 are incorporated below.

The following has been added after line 16.

"*The following sections discuss in detail how the results from TF-HTAP2 have been incorporated into the parameterisation.*"

L25. It is not clear how the parameterization has been improved.

The sentence on Line 25 has been changed to:

"*The parameterisation of Wild et al., (2012) was used to calculate new baseline $O_3$ concentrations in 2010 for use in this version of the parameterisation and for comparison to the TF-HTAP2 multi-model mean.*"

L24-34. In Table 1 the emission differences are averaged for both the MACCity and EDGAR inventories? It is not clear why the authors did not average the emission differences as used in the TF-HTAP2 models versus those in the TF-HTAP1 models. It is unclear how or why internal consistency (whatever that means) should be relevant here. The actual difference in emissions would seem to be more relevant in comparing the change in O3.

The TF-HTAP1 models each used their own emissions as input for experiments (Fiore et al., 2009) and therefore there is no consistent baseline to compare to the prescribed TF-HTAP2 emissions. This is mentioned in the manuscript on P5 Line 19 and Line 29. Therefore the EDGAR and MACCity emission inventories were chosen to provide an emission change between year 2000 and 2010 using consistent datasets in both time periods. The change in precursor emissions between 2000 and 2010 was then used to provide new baseline ozone concentrations in 2010 for use in the parameterisation (see point below). The following text on P5 Line 26 to 30 has been amended to make this clearer.

"*The mean fractional change in $NO_x$, CO and NMVOC emissions between 2000 and 2010 across the TF-HTAP1 source regions from two different emission inventories, MACCity (Granier et al., 2011) and EDGARv4.3.1 (Crippa et al., 2016), was used (Table 1), as the use of a specific emission inventory was not prescribed for TF-HTAP1 experiments (See – Fiore et al., 2009).*"

"*The MACCity and EDGAR inventories provide a consistent set of emissions in 2000 and 2010, enabling the change in emissions between future and historical time periods to be explored.*"

L35 - how many models contributed to the parameterized ozone response? -In general it is not really clear what was done here. Were the emission differences in table 1 used to compute the parameterized change in ozone in table 2? Was the parameterized response from each model computed separately to give the standard deviation in the parameterized response?

The parameterisation of Wild et al., (2012) was used to calculate new baseline $O_3$ concentrations in 2010 (Table 2) using the emission differences in Table 1 (see lines 24 to 34.) This version of the parameterisation used 14 models from the TF-HTAP1 experiments and computed responses individually which were then averaged to give the multi-model mean response and standard deviation from the parameterisation. Line 35 has been amended to make these points clearer.

"*The parameterised surface ozone response in 2010 was calculated using the method of Wild et al., (2012), based on the individual response of 14 TF-HTAP1 models using the fractional emission changes in Table 1. The parameterised ozone response across the ….*"

L37-38. "similar to the TF-HTAP2 multi-model mean values". This seems a little misleading as the parameterized responses are also similar to the TF-HTAP1 values. It would be more insightful to quantify the extent to which the parameterization quantifies the changes between TF-HTAP1 and TF-HTAP2.

The reviewer makes a valid point that the TF-HTAP1 and TF-HTAP2 results are similar and the remaining part of this paragraph goes onto make the point that the larger range in ozone responses for both TF-HTAP1 and TF-HTAP2 dominates the uncertainty in ozone concentrations and is much larger than any response from changing emissions between 2000 and 2010. Therefore any predictions using the parameterisation for ozone response between 2000 and 2010 will still have an uncertainty associated with it mainly due to the large spread in model responses.

Changed Line 27 to 38 to the following:

"*Table 2 shows that the $O_3$ concentrations from the parameterisation (H-P) are within the spread of the individual model values from TF-HTAP2 (H-2), represented by one standard deviation, over most of the receptor regions.*"

L5. "adjusted". I assume the authors explain how the emissions are adjusted below.

The emissions were not adjusted here. The $O_3$ response fields from the European, North American, East Asian and South Asian source regions of TF-HTAP1 models were reapportioned so that a larger number of models (14 in total) than available from TF-HTAP2 are able to represent the equivalent TF-HTAP2 source regions. P6 Lines 8 to 10 specifically mentions the adjustment of $O_3$ response fields with the rest of section 2.3.2 explaining in more detail how this source region adjustment of TF-HTAP1 models was performed.

L11-21. Both the source and receptor regions are changed between HTAP1 and HTAP2. It is unclear from the description here how you discretize the response in the HTAP1 models into both smaller source regions and smaller receptor regions. The discussion in 2.3.2 seems to only concern the source region adjustment. Sections 2.3.2 and 2.3.3 should be clarified as to the exact procedure used.

The source region adjustment has been described here as this is the only part of the parameterisation that requires adjusting. Additional simulations were performed with HadGEM2-ES (which participated in TF-HTAP2) using the TF-HTAP1 source regions (Europe, North America, East Asia, South Asia) to inform and evaluate this source region adjustment. The methodology for this is described in section 2.3.2.

No specific adjustment is required for the receptor regions, as these can be defined arbitrarily based on the global distribution of ozone responses generated in both TF-HTAP1 and TF-HTAP2 studies. Here we have defined the receptor regions to match the source regions used in TF-HTAP2 for consistency. This is referred to on P7 Line 19 which notes 'Output is provided on a standard grid to facilitate the calculation of $O_3$ responses over any selected receptor regions.' An additional comment referring to the definition of receptor regions has been included on P6 Line 25.

*"Here, receptor regions are defined in accordance with those in TF-HTAP2 (16 in total), although it is possible to define any required receptor regions using the global distribution of O₃ responses."*

-L16-23. This seems to be largely a repeat of what is said above.

See changes made to relevant text in next response.

-L17 "significant improvements". In what way? Are more models are used, or are the source-receptor regions are better defined, or do the authors feel the parameterization itself has been improved in some fundamental way? The improved parameterization is again mentioned on page 9, line 19 (and probably elsewhere). Please be explicit on how exactly the parameterization is improved.

The underlying approach used in the parameterisation remains the same as Wild et al., (2012) but improvements have been made to the input and output. These improvements are described in the remaining lines of the paragraph. The text on P7 lines 16 to 22 have been replaced with the following to try and make the improvements clearer.

*"The original parameterisation developed by Wild et al., (2012) was based on the surface O₃ response to 20% continental-scale emission perturbations from TF-HTAP1 for 2001. We have adopted the same approach but have made a number of major improvements: updating the base year to 2010, included additional models from TF-HTAP2, extending the number of source regions to 14, and generating three-dimensional O₃ responses to permit calculation of tropospheric O₃ burden and O₃ radiative forcing for any scenario. To test and verify the improved parameterisation, additional simulations have been conducted with HadGEM2-ES, which are discussed in the following sections."*

-L31 "is working well". This is a little hard to tell from the figures. It would be valuable to show the percentage error as a function of month for the two responses.

We have chosen to present absolute changes as this is the measure that matters most, and is also easiest for the reader to interpret. Percentage errors exaggerate differences where the underlying O₃ responses are very small. Therefore we think it is best to focus on absolute errors (ppb) as that is discussed throughout the manuscript. For the reviewers convenience we have also computed the relative errors in the table below from the data used to plot Figure 2. The relative errors are largest, in transitioning from the winter titration regime to the spring/summer ozone production season. Generally errors are below 20% apart from a few select months (March, April, September and October).

| Emission Reduction | | Jan | Feb | Mar | Apr | May | Jun | Jul | Aug | Sep | Oct | Nov | Dec |
|---|---|---|---|---|---|---|---|---|---|---|---|---|---|
| 20% | Param | 0.700 | 0.598 | 0.279 | -0.192 | -0.571 | -0.743 | -0.771 | -0.600 | -0.211 | 0.190 | 0.519 | 0.635 |
| | HG | 0.597 | 0.642 | 0.234 | -0.071 | -0.542 | -0.734 | -0.724 | -0.570 | -0.265 | 0.225 | 0.453 | 0.584 |
| | Error | -0.103 | 0.045 | -0.046 | 0.121 | 0.029 | 0.009 | 0.047 | 0.030 | -0.054 | 0.035 | -0.067 | -0.051 |
| | % err | -17% | 7% | -20% | **-169%** | -5% | -1% | -7% | -5% | 20% | 16% | -15% | -9% |
| 50% | Param | 1.574 | 1.342 | 0.608 | -0.553 | -1.520 | -1.964 | -2.038 | -1.601 | -0.611 | 0.382 | 1.158 | 1.427 |
| | HG | 1.563 | 1.617 | 0.451 | -0.396 | -1.653 | -2.124 | -2.096 | -1.689 | -0.952 | 0.415 | 1.114 | 1.515 |
| | Error | -0.011 | 0.275 | -0.156 | 0.157 | -0.133 | -0.160 | -0.058 | -0.088 | -0.341 | 0.033 | -0.043 | 0.089 |
| | % err | -1% | 17% | -35% | **-40%** | 8% | 8% | 3% | 5% | 36% | 8% | -4% | 6% |
| 75% | Param | 2.141 | 1.824 | 0.798 | -0.922 | -2.395 | -3.080 | -3.195 | -2.528 | -1.021 | 0.457 | 1.561 | 1.940 |
| | HG | 2.402 | 2.351 | 0.388 | -1.011 | -2.996 | -3.673 | -3.632 | -2.987 | -1.998 | 0.278 | 1.569 | 2.309 |
| | Error | 0.261 | 0.526 | -0.410 | -0.090 | -0.601 | -0.594 | -0.438 | -0.459 | -0.977 | -0.179 | 0.008 | 0.369 |
| | % err | 11% | 22% | **-106%** | 9% | 20% | 16% | 12% | 15% | **49%** | **-64%** | 1% | 16% |

Page 8.

-The results over South Asia are really quite strange as the parameterization based on HadGEM2 fits the multi-model parameterization. The authors seem to be arguing on page 8 and 9 that this is a difficult region to simulate and that perhaps it is not surprising that the parameterization based on HadGEM with the large titration might not be able to simulate this region accurately. However, this seems to be only half of the story…. Why does the parameterization based specifically on HadGEM match the multi-model parameterization? And does the multi-model parameterization capture the multi-model response in this region?

The reviewer is correct to point out that the results over South Asia are different from that over other regions and highlights a particular limitation in the parameterised approach. The parameterisation is based on ozone response fields from 20% emission reduction experiments, but in the ECLIPSE CLE scenario there is a ~70% increase in NOx emissions over South Asia. The ozone responses generated by a 20% emission reduction show ozone reductions over clean, marine regions, but are close to zero over the continent (see Figure R1 below). Table 5 shows the parameterisation matches the TF-HTAP2 multi-model mean response well over South Asia for a 20% emission reduction.

For the 70% increase in NOx emissions over South Asia the parameterisation generates an increase in ozone over South Asia in January (Figure R2), scaled from the 20% emission change (Figure R1), whereas the HadGEM2-ES modelled response is a decrease in ozone (Figure R3). The simple scaling within the parameterisation is not able to simulate the shift in chemical environment over South Asia from production to loss that is associated with this large increase in emissions. This is because it is based on ozone distributions from only two model runs, and is thus not able to represent the strong non-linearities that may arise from large emission changes over regions which are close to maximum ozone production in current conditions. We note that the errors would be greater if we used a linear assumption. Future developments could attempt to address this problem by including additional simulations that would allow a full characterisation of the ozone response over a much wider range of emission changes.

[Figure]

**Figure R1** – January ozone response in HadGEM2-ES to a 20% emission reduction over South Asia of all anthropogenic precursor emissions (NOx, CO, NMVOCs)

[Figure]

**Figure R2** – January ozone response from the parameterisation to the emission changes in the ECLIPSE CLE 2030 scenario, relative to 2010

[Figure]

**Figure R3** – January ozone response from HadGEM2-ES to the emission changes within the ECLIPSE CLE 2030 scenario, relative to 2010

The HadGEM2-ES simulation using emission changes from the ECLIPSE CLE scenario in 2030 was conducted to evaluate the parameterisation against a global emission perturbation and did not form part of the TF-HTAP2 set of experiments. No other model results are available using this scenario to identify if the results over South Asia are specific to HadGEM2-ES. Discussion of this occurs on Page 9, line 10.

The following sections of the text on P8 Lines 31 to 40 have been amended to improve the description of the result over South Asia, with the above figures included in the supplementary material:

*"The large increase in emissions causes the chemical environment in HadGEM2-ES in January to shift from $O_3$ production to that of titration (Figure S3). The parameterisation is not able to represent this shift (Figure S4) as it is based on a single ozone response to a 20% emission reduction (Figure S5) and is unable to capture the strongly non-linear transition into a net ozone titration regime. This is a smaller problem over North America, Europe or East Asia, as wintertime titration regimes are already present over these regions. This effect seen over South Asia highlights a weakness in the parameterised approach in representing strongly non-linear chemical regimes where there are large emission changes, although we note that the errors would be worse if a linear scaling was used."*

-L25-26: "compare well with ACCMIP multi-model means for intermediate emission scenarios…" What about RCP4.5? Isn't this a intermediate emission scenario? The comparison from RCP4.5 does not look that good.

See response to Point 2 above where a revised comparison with ACCMIP models is presented.

**Response to Referee 2**

The manuscript by Turnock et al. presents an updated version of earlier work done by Wild et al. (2012), who constructed a parameterisation for calculating the response of tropospheric ozone to changes in precursor emissions (and methane abundances). The parameterisation reduces the need to run an ensemble of computationally expensive global models of atmospheric chemistry in order to explore the effects of different emission scenarios on the abundance of tropospheric ozone, and the need to run multiple model experiments to determine the influences of emissions from multiple source regions on individual receptor regions. As such, this is a tool for rapid assessment of alternative emission control policies, but does not replace global atmospheric chemistry models, since it does not represent the nonlinearities of tropospheric ozone chemistry, or the influence of future climate changes on tropospheric ozone. As an update to Wild et al. (2012), I expect that this paper will be widely used.

Unfortunately I found much of the description of the method to be vague and confusing. This should be improved before the paper is published. Also, the authors could do more to compare the predictions of their parameterised ozone with actual simulations from global atmospheric chemistry models. More details are given below:

The description of parameterisation has been improved by changes made to Section 2.1. Please see the response to point 1 for reviewer 1 above for the full text with the relevant sections included again in response to the specific points below.

Page 3, lines 5-7: Ignoring changes in future ozone due to climate change is an important limitation of this study. Here it would be appropriate to give a short summary of the expected changes in surface ozone due to climate change, to provide the reader with more information about this limitation.

The following text on the impact of climate change on surface ozone has been added on page 3 to the end of line 7:

*"Future climate change is expected to alter surface concentrations of ozone through changes to meteorological variables such as temperature, precipitation, water vapour, clouds, advection and mixing processes (Doherty et al., 2017)."*

Page 3, line 13: There appears to be a typo in the last sentence of this paragraph.

Removed extra $O_3$ from sentence so it becomes the following:

"*Section 5 uses the same future emission scenarios to predict future tropospheric $O_3$ burden and radiative forcing.*"

Page 4, line 5: Why is f replaced with g? Is this just a difference in terminology between Wild et al. (2012) and this study? Or something else?

As the parameterisation is based on that in Wild et al., (2012) the same terminology was used for consistency purposes. Wild et al., (2012) defined the linear scale factor as *f* for CO and NMVOC emission perturbations whereas the non-linear scale factor is represented by *g* for perturbations to NOx emissions and $CH_4$ abundances. Equation 2 has been replaced with Equations 1 to 5, shown in the response to point 1 of reviewer 1, to make the notation clearer. The revised equation 4 is shown below (for more details see the response to point 1 in reviewer).

$$f_{ij} = 0.95r_{ij} + 0.05r_{ij}^2 \quad Scaling\ accounting\ for\ reduced\ O_3\ increases\ from\ NO_X\ and\ CH_4 \qquad (4)$$

P4, Line 4 - 6 has been replaced with the following:

*"To account for non-linear behaviour of surface $O_3$ to $NO_x$ emission changes, a quadratic scaling factor (Eq. 4) is used, based on additional simulations of surface $O_3$ response over a larger range of emission perturbations in Wild et al., (2012)."*

Page 3, line 9: Where does "2f -g" come from? This appears to come out of nowhere.

This expression represents a reversal in curvature for NOx emission reductions in titration regimes (where increased emissions lead to reduced ozone) and is described in more detail in Wild et al., 2012. However, we have revised the notation used here in response to the comments from both reviewers (see new Equations 1 to 5 in response to point 1 of reviewer 1). The revised equation 5 is shown below.

$$f_{ij} = 1.05r_{ij} - 0.05r_{ij}^2 \quad Scaling\ for\ titration\ regimes\ where\ decreasing\ NO_X\ increases\ O_3 \qquad (5)$$

P4 Lines 8 to 11 have been amended as follows:

*"For the special case of source regions that are under titration regimes, where a reduction in NOx emissions may lead to an increase in $O_3$, the curvature of the response is reversed for NOx emission decreases (Eq. 5), as described in Wild et al., (2012). A linear scale factor is used for emission increases under these conditions (Eq. 3)."*

Page 3, line 10: Much more explanation is needed here. Which model simulations? Which year? Why is the spatial extent of the titration regimes important? How is the magnitude defined? A lot of very important information appears to have simply been left out.

P4 Line 10 has been altered to the reflect the fact that the parameterisation is based on the difference between two model simulations and is unable to calculate future changes to chemical regimes.

*"The spatial extent of ozone titration is assumed constant as the parameterisation is based on differences between two model simulations and is therefore unable to represent any future changes in chemical regime."*

Page 3, line 14: It's fascinating to read that the response of surface ozone to methane, a well-mixed gas with a lifetime measured in years, could be similar to the response to NOx, which has a lifetime on the order of hours. In what way is the response "similar"? More explanation is needed here.

As we state in the manuscript, the non-linear behaviour of the ozone response to $CH_4$ is similar to that of NOx emissions but not the magnitude of the response. The same quadratic function is therefore used within the parameterisation to scale $O_3$ responses from changes in $CH_4$ abundances and NOx emissions. However, we appreciate the confusion here so have re-written line 14 to make this clearer.

*"The surface $O_3$ response to changes in global $CH_4$ abundance shows a similar degree of non-linearity as that from changes in NOx emissions (Wild et al., 2012). Therefore, the non-linear scale factor (Eq. 4) is also used to represent the $O_3$ response to changes in $CH_4$ abundances."*

Equation 2: How are the coefficients in the calculation of g determined?

The coefficients for g are those previously determined by additional simulations in (Wild et al., 2012). Page 4 Line 4 has been amended to the following (see point 1 for reviewer 1):

*"To account for non-linear behaviour of surface $O_3$ to $NO_x$ emission changes, a quadratic scaling factor (Eq. 4) is used, based on additional simulations of surface $O_3$ response over a larger range of emission perturbations in Wild et al., (2012)."*

Page 8, line 40: There is no need to mention titration a second time.

We state here that the ozone response over South Asia from HadGEM2-ES simulations could be due to a shift to an ozone titration regime in the model which is not represented in the parameterisation, and thus explains some of the discrepancy between them. The important part here is the shift in regime, which has not been discussed previously and we feel is important to keep in the manuscript.

Section 3.3.1: Can you compare the predictions of the parameterisation using the RCP scenarios with the actual global model runs done in the ACCMIP exercise? See Young et al. (2013) for some examples. This comparison would really help the reader to understand more about how well the parameterisation is doing in comparison with the global models. Perhaps this comparison could be added to Fig. 5. Also, why isn't this part of Section 4? ("Future Surface Ozone Predictions")

The comparison with the RCPs provides an evaluation of future predictions of the parameterisation as ozone projections for these scenarios have already been made previously using the parameterisation of Wild et al., (2012). The parameterised predictions of global surface ozone, global tropospheric burden and ozone radiative forcing in 2030 from each of the RCPs have now been compared to that from the ACCMIP models in the new version of Table 9 (see response to point 2 of reviewer 1). However, it should be noted that changes to ozone concentrations in the ACCMIP models will have additional contributions from climate change and stratospheric ozone recovery, which are not represented in the parameterisation.

Table 9 has been amended as described in the response to point 2 for reviewer 1 above (Table 10 in revised manuscript) and now includes a comparison to surface ozone from the ACCMIP models.

The following text has been included to discuss the comparison to ACCMIP models (see response to point 2 for reviewer 1 for more details).

P13 Lines 22 to 30 have been amended to:

*"In comparison to the ACCMIP multi-model mean, the predicted changes between 2000 to 2030 in both global annual mean surface $O_3$ and global $O_3$ burden from the parameterisation are within the range of the ACCMIP multi model responses (+/- 1 standard deviation) for all the CMIP5 RCPs (Table 10). The predictions of $O_3$ radiative forcing in 2030 from the parameterisation across all the RCPs, when the influence of climate change is anticipated to be small, are also consistent with those from ACCMIP. The sign and magnitude of change in global $O_3$ burden and $O_3$ radiative forcing with the parameterisation for RCP6.0 is different from the ACCMIP results but is still within the range of model responses, which is the largest for this scenario. The comparison with ACCMIP results shows that the parameterisation is able to reproduce changes in global $O_3$ burden and $O_3$ radiative forcing on near-term timescales, when the influence of climate change is small."*

Page 10, line 8: "successive emission increases" of what magnitude? In what succession?

Each additional increment in emissions (an amount corresponding to 20% of current emissions) gives a 10% smaller ozone response than the previous one. This reflects the nonlinearity described by the quadratic expression that is given.  Line 8 has been re-worded to the following:

*"For simplicity the parameterisation used the same non-linear scaling factor as for $NO_x$ emissions (Eq. 4 ), which represents a 10% smaller response for successive 20% emission increases."*

Equation 3: The coefficients appear to have changed since Equation 2. Why? How are they calculated? This seems like something for Section 2.

This section uses results from the models in TF-HTAP2 that conducted methane perturbation experiments to see if the coefficients derived for the quadratic function in Wild et al., (2012) are still valid (P10, line 8). This used the same method as in Wild et al., (2012) to derive the coefficients for the TF-HTAP2 models, shown in equation 3. Since equation 3 and 2 are similar and within the level of uncertainty it was decided for consistency purposes to retain the same coefficients of the quadratic function used in Wild et al., (2012) (stated on P10 Lines 11 to 12). As this was a comparison of TF-HTAP2 results to TF-HTAP1 and evaluation of the existing parameterisation it was decided to put this discussion into Section 3 – Testing and Validation. To avoid further confusion we have removed equation 3 from the manuscript.

*" (3)"*

P10 Line 11 has been amended as follows to reflect this change:

*"We find a slightly larger sensitivity, with both models yielding a 12.6% smaller surface $O_3$ response for an increase in $CH_4$ than a decrease ."*

Reference to equation 3 is also removed from P10 L17-19:

*"To enable a direct comparison with TF-HTAP1 results, the $O_3$ response from the CH4DEC and CH4INC experiments in TF-HTAP2 are scaled to represent the response from a 20% reduction in $CH_4$ abundances. An adjustment factor is calculated based on the global mean difference between the TF-HTAP2 $O_3$ response in each experiment and that of an equivalent 20% reduction in $CH_4$ abundance , resulting in a factor of 1.557 for CH4DEC and -1.256 for CH4INC."*

Page 10, line 33: What are the "appropriate fields"?

P10, line 33 have been modified to include the appropriate fields.

*"Table 7 summarises the calculated $CH_4$ lifetime and feedback factors for the two TF-HTAP2 models that provided $CH_4$ chemical loss rates."*

Section 4.1: Can you compare the predictions of the parameterisation using the ECLIPSE scenarios with the actual global model runs done in the ECLIPSE exercise? See Stohl et al. (2015, DOI: 10.5194/acp-15-10529-2015) and work citing that paper for examples.

A different version of the ECLIPSE emissions inventory has been used as input to the models for the ECLIPSE project (Version 4 and 5) than was used with the parameterisation here (Version 5a). Different future emissions pathways exist in these versions of the emission inventory and this hinders a direct comparison between the models and the parameterisation. Therefore, we think it is not feasible to provide a direct comparison between the ECLIPSE models and the parameterisation here. Whereas output from the ACCMIP models was available for the same set of CMIP5 future RCPs, making a direct comparison between these models and the parameterisation more appropriate.

However, the percentage change difference in surface ozone concentrations between the mitigation and current legislation scenario in 2050 is presented in Table 4 of Stohl et al., (2015). This shows a multi-model reduction of surface ozone concentrations over Europe, China, India and the United States of between 13 to 20% for the mitigation scenario. Whilst not directly comparable to our study in terms of scenarios used or receptor regions we calculate a similar reduction in future surface ozone concentrations of approximately 20% over Europe, North America and East Asia for the ECLIPSE MTFR scenario in 2050 (relative to CLE) and ~30% for South Asia.

**Response to Short Comment by R. Van Dingenen**

In general this is a well-written and very useful paper that addresses relevant policy issues.

As a possible user of the ozone precursor source-receptor relations, I would like to make some suggestions that would improve the readability of the paper and create the possibility for the scientific community to replicate the results.

Eq. 1: the same variable symbol (deltaO3) is used at left and right-hand side of the equation, while they have different meanings. The same observation can be made for Eq. 2 where e.g is written fij = 2fij - gij; suggest to use a different symbol at the left hand side.

We appreciate the possible confusion here, and have adjusted equation 1 by adding the subscripts e (for emissions) and m (for methane) to the delta-O3 terms on the right-hand side to distinguish the delta-O3 terms from each other. We have left the overall form of the equation as it is, for consistency with Wild et al., (2012). We have revised the notation in Eq.2 to clarify the expressions in response to the comments of the reviewers (see the response to point 1 for reviewer 1).

Eq. 1 expresses deltaO3 as response to the sum of an emission change (for NOx, CO and NMVOCs), and an abundance change in CH4. For the user, using emission changes for all precursors would make more sense. Isn't it possible, from the box model mentioned in section 3.2, and using a feedback factor, to relate a change in abundance to a change in emissions? Why not normalise the source-receptor responses by the emission strength? It would be useful to emphasise the time scale of the CH4 responses and how to deal with this in such a parametrised approach.

The parameterisation is based on simulations from HTAP1 and HTAP2 that used a change in global methane abundance to simulate an $O_3$ response and so the parameterisation is based on methane abundance. It would be possible to extract a $CH_4$ emission response based on the prescribed $CH_4$ abundance change (e.g. Meinshausen et al., 2011; Holmes et al., 2013), but this would be different for each model, and including this adds an additional layer of complexity to the parameterisation.

Future developments could include some sort of emissions to abundance conversion as a post processing step.

It's not clear why paragraph 3.1 is named 'Scaling Factors'

Change section title to "*Limits of Linear Scaling*".

Page 10, line 12: 'the same scaling factor', is not clear if 'same' refers to using the same as in HTAP1, or using the same (new) factor for CH4 and NOx. So, Eq. 3: is this now the scaling factor replacing the 0.95f+0.05fˆ2 from HTAP1 both for NOx and CH4?

Changed Line 12 to:

"… *same representation of non-linearity for both NOx and CH$_4$ (Eq. 4), as used in Wild et al., (2012)*"

Figures 7 and 8 (and similar in SI): does the ozone trend from CH4 include the transient effect of the 12y perturbation response time? How can Eq. 3 be applied (for CH4) to obtain this trend? The figures show the change in ozone relative to year 2010; does it include the time-lagged impact of CH4 emissions before that date? I would appreciate having the box model for CH4 better documented.

The contribution to the ozone trend from changes in methane reflects the effects of the change in methane abundance alone, and is the equilibrium response (short and long term). No transient effects are considered in the parameterisation, and this is another argument for the simplicity of basing it on abundance rather than emissions. The abundance is calculated using the methods and expressions given in Meinshausen et al., (2011) and Holmes et al., (2013) and no new aspects have been introduced for the purposes of this study.

**References**

Crippa, M., Janssens-Maenhout, G., Dentener, F., Guizzardi, D., Sindelarova, K., Muntean, M., Van Dingenen, R. and Granier, C.: Forty years of improvements in European air quality: regional policy-industry interactions with global impacts, Atmos. Chem. Phys., 16(6), 3825–3841, doi:10.5194/acp-16-3825-2016, 2016.

Doherty, R. M., Heal, M. R. and O'Connor, F. M.: Climate change impacts on human health over Europe through its effect on air quality, Environ. Heal., 16(S1), 118, doi:10.1186/s12940-017-0325-2, 2017.

Fiore, A. M., Dentener, F. J., Wild, O., Cuvelier, C., Schultz, M. G., Hess, P., Textor, C., Schulz, M., Doherty, R. M., Horowitz, L. W., MacKenzie, I. A., Sanderson, M. G., Shindell, D. T., Stevenson, D. S., Szopa, S., Van Dingenen, R., Zeng, G., Atherton, C., Bergmann, D., Bey, I., Carmichael, G., Collins, W. J., Duncan, B. N., Faluvegi, G., Folberth, G., Gauss, M., Gong, S., Hauglustaine, D., Holloway, T., Isaksen, I. S. A., Jacob, D. J., Jonson, J. E., Kaminski, J. W., Keating, T. J., Lupu, A., Manner, E., Montanaro, V., Park, R. J., Pitari, G., Pringle, K. J., Pyle, J. A., Schroeder, S., Vivanco, M. G., Wind, P., Wojcik, G., Wu, S. and Zuber, A.: Multimodel estimates of intercontinental source-receptor relationships for ozone pollution, J. Geophys. Res. Atmos., 114(4), 1–21, doi:10.1029/2008JD010816, 2009.

Granier, C., Bessagnet, B., Bond, T. C., D'Angiola, A., Denier van der Gon, H., Frost, G. J., Heil, A., Kaiser, J. W., Kinne, S., Klimont, Z., Kloster, S., Lamarque, J.-F., Liousse, C., Masui, T., Meleux, F., Mieville, A., Ohara, T., Raut, J.-C., Riahi, K., Schultz, M. G., Smith, S. J., Thompson, A., Aardenne, J., Werf, G. R. and Vuuren, D. P.: Evolution of anthropogenic and biomass burning emissions of air pollutants at global and regional scales during the 1980–2010 period, Clim. Change, 109(1–2), 163–190, doi:10.1007/s10584-011-0154-1, 2011.

Holmes, C. D., Prather, M. J., Søvde, O. A. and Myhre, G.: Future methane, hydroxyl, and their uncertainties: key climate and emission parameters for future predictions, Atmos. Chem. Phys. Atmos. Chem. Phys., 13, 285–302, doi:10.5194/acp-13-285-2013, 2013.

Meinshausen, M., Raper, S. C. B. and Wigley, T. M. L.: Emulating coupled atmosphere-ocean and carbon cycle models with a simpler model, MAGICC6 - Part 1: Model description and calibration, Atmos. Chem. Phys., 11(4), 1417–1456,

doi:10.5194/acp-11-1417-2011, 2011.

Stevenson, D. S., Young, P. J., Naik, V., Lamarque, J.-F., Shindell, D. T., Voulgarakis, A., Skeie, R. B., Dalsoren, S. B., Myhre, G., Berntsen, T. K., Folberth, G. A., Rumbold, S. T., Collins, W. J., MacKenzie, I. A., Doherty, R. M., Zeng, G., van Noije, T. P. C., Strunk, A., Bergmann, D., Cameron-Smith, P., Plummer, D. A., Strode, S. A., Horowitz, L., Lee, Y. H., Szopa, S., Sudo, K., Nagashima, T., Josse, B., Cionni, I., Righi, M., Eyring, V., Conley, A., Bowman, K. W., Wild, O. and Archibald, A.: Tropospheric ozone changes, radiative forcing and attribution to emissions in the Atmospheric Chemistry and Climate Model Intercomparison Project (ACCMIP), Atmos. Chem. Phys., 13(6), 3063–3085, doi:10.5194/acp-13-3063-2013, 2013.

Stohl, A., Aamaas, B., Amann, M., Baker, L. H., Bellouin, N., Berntsen, T. K., Boucher, O., Cherian, R., Collins, W., Daskalakis, N., Dusinska, M., Eckhardt, S., Fuglestvedt, J. S., Harju, M., Heyes, C., Hodnebrog, Ø., Hao, J., Im, U., Kanakidou, M., Klimont, Z., Kupiainen, K., Law, K. S., Lund, M. T., Maas, R., MacIntosh, C. R., Myhre, G., Myriokefalitakis, S., Olivié, D., Quaas, J., Quennehen, B., Raut, J.-C., Rumbold, S. T., Samset, B. H., Schulz, M., Seland, Ø., Shine, K. P., Skeie, R. B., Wang, S., Yttri, K. E. and Zhu, T.: Evaluating the climate and air quality impacts of short-lived pollutants, Atmos. Chem. Phys., 15(18), 10529–10566, doi:10.5194/acp-15-10529-2015, 2015.

Wild, O., Fiore, A. M., Shindell, D. T., Doherty, R. M., Collins, W. J., Dentener, F. J., Schultz, M. G., Gong, S., Mackenzie, I. A., Zeng, G., Hess, P., Duncan, B. N., Bergmann, D. J., Szopa, S., Jonson, J. E., Keating, T. J. and Zuber, A.: Modelling future changes in surface ozone: A parameterized approach, Atmos. Chem. Phys., 12(4), 2037–2054, doi:10.5194/acp-12-2037-2012, 2012.